# MAC-AMP: A Closed-Loop Multi-Agent Collaboration System for Multi-Objective Antimicrobial Peptide Design

**Gen Zhou[1]\*, Sugitha Janarthanan[2]\*, Lianghong Chen[1], Pingzhao Hu[1,2]†**
[1]Department of Computer Science, Western University, London, ON, Canada
[2]Department of Biochemistry, Western University, London, ON, Canada

## Abstract

To address the global health threat of antimicrobial resistance, antimicrobial peptides (AMPs) are being explored for their potent and promising ability to fight resistant pathogens. While artificial intelligence (AI) is being employed to advance AMP discovery and design, most AMP design models struggle to balance key goals like activity, toxicity, and novelty, using rigid or unclear scoring methods that make results hard to interpret and optimize. As the capabilities of Large Language Models (LLMs) advance and evolve swiftly, we turn to AI multi-agent collaboration based on such models (multi-agent LLMs), which show rapidly rising potential in complex scientific design scenarios. Based on this, we introduce **MAC-AMP**, a closed-loop multi-agent collaboration (MAC) system for multi-objective AMP design. The system implements a fully autonomous simulated peer review-adaptive reinforcement learning framework that requires only a task description and example dataset to design novel AMPs. The novelty of our work lies in introducing a closed-loop multi-agent system for AMP design, with cross-domain transferability, that supports multi-objective optimization while remaining explainable rather than a 'black box'. Experiments show that MAC-AMP outperforms other AMP generative models by effectively optimizing its AMPs for multiple key molecular properties, demonstrating exceptional results in antibacterial activity, AMP likeliness, toxicity compliance, and structural reliability.

## 1 Introduction

Although new antibiotics are still being developed to treat bacterial infections, antimicrobial resistance (AMR) remains a critical challenge, one that has caused a systemic crisis in global public health. AMR occurs when bacteria evolve mechanisms that reduce or eliminate the effectiveness of drugs designed to kill them, making bacterial infections harder to treat. In 2021 alone, bacterial AMR directly caused approximately 1.14 million deaths and was associated with approximately 4.71 million deaths. In addition, it is estimated that between 2025 and 2050, AMR will directly lead to over 39 million deaths across all ages and will be associated with 169 million deaths (Naghavi et al., 2024). Antimicrobial peptides (AMPs) are naturally occurring short chains of amino acids that are part of the innate immune system and serve as the natural defence against a broad range of microbes in many living organisms. They are being explored to combat AMR due to their impressive properties, including broad-spectrum activity, diverse mechanisms of action, and higher resistance barriers compared to traditional antibiotics. However, they still face bottlenecks such as risk of toxicity and hemolysis, insufficient stability and bioavailability *in vivo*, and limitations in manufacturability and cost (Bucataru & Ciobanasu, 2024; Min et al., 2024).

Recently, scientists have turned to artificial intelligence (AI) models to design AMPs. Over the past couple of years, AI-driven AMP discovery has expanded from retrieval and screening to include generation and optimization (Pirtskhalava et al., 2021). Recent AMP generative and discriminative models have achieved encouraging progress on public benchmarks, and some studies have validated

---

*Equally Contributed
†Contact Author: phu49@uwo.ca

several *in vitro* active candidates. However, most AI-driven AMP design models face limitations. First, most optimize solely for activity, which tends to produce AMPs with undesirable molecular properties, limited synthesizability, and restricted novelty (Van Oort et al., 2021; Tucs et al., 2020). Those that employ multi-objective optimization are often unstable, as static weighting or thresholding can cause reward hacking or diversity collapse (Abels et al., 2019). In addition, their outputs are usually scattered scores or text, which are hard to convert into clear, reproducible learning signals for stable reinforcement-style optimization (Van Kempen et al., 2024; Wu et al., 2022; Guan et al., 2025). To combat these gaps, multi-agent collaboration (MAC) systems are being explored. MAC systems emphasize solving complex tasks through division of labour, communication, and collaboration among interacting autonomous agents. Large Language Models (LLMs) are now being used as flexible interfaces and reasoning agents within MAC systems, making LLM-based MAC systems very popular. A review published in late 2024 on LLM-based MAC systems highlighted significant advances in complex problem-solving and world simulation, demonstrating the effectiveness of research frameworks in which collaborative agents interact to execute sophisticated scientific and engineering workflows across diverse domains (Guo et al., 2024). However, a critical limitation of current MAC systems is that their outputs are typically in the form of natural language or heterogeneous scores, lacking reproducible training signals suitable for model optimization. A recent model, Eureka, has shown that LLMs that generate and self-improve their own reward code can significantly boost reinforcement learning (RL) performance, but it has yet to be integrated into MAC systems (Ma et al., 2023). In addition, recent LLM-based MAC systems are largely open-loop, as they converse, call tools, and often involve human-in-the-loop operation (Wu et al., 2024). As a result, downstream optimization often relies on trial-and-error prompt iteration or *ad-hoc* fine-tuning rather than principled closed-loop learning (Song et al., 2024).

To address these gaps, we introduce MAC-AMP, the first closed-loop MAC system for AMP design. Unlike prior AMP generators that treat design as a single-model sequence optimization task, we recast AMP design as a coordinated multi-agent problem and propose a general, end-to-end pathway that translates a user's design request into novel, multi-objective–optimized AMPs. MAC-AMP integrates four modules: (1) a Property Prediction module that applies specialized scoring tools to evaluate AMPs on activity, safety, stability, and novelty; (2) an AI-Simulated Peer Review module, in which specialized agents synthesize these evaluations into structured, multi-criteria consensus rather than relying on isolated scalar scores; (3) an RL Refinement module that translates agent consensus into machine-actionable reward functions, replacing free-text or ad hoc weighting heuristics; and (4) a Peptide Generation module that closes the loop by dynamically and transparently adapting the training objective during peptide design, enabling stable optimization under conflicting biological constraints. The key architectural novelties of MAC-AMP are:

1. A **fully autonomous, closed-loop multi-agent system** that converts AMP-specific evaluations into executable RL reward signals, which establishes a real-time feedback cycle for continuous design, critique, and optimization.

2. **Stepwise explainability and auditability** via transparent logs, replay traces, and consensus-aware decision tracking across all agents, overcoming black-box limitations of AI models, introducing explainability, and enabling error localization and systematic correction.

3. **Native support for multi-objective AMP design**, balancing antibacterial activity, structural stability, toxicity, and other constraints through structured agent consensus rather than manual or static weighting schemes.

4. A **domain-agnostic framework** that supports transferability beyond AMP generation.

Overall, MAC-AMP sets a new benchmark for generative AMP design, surpassing existing models in antibacterial activity, toxicity reduction, and structural reliability while maintaining comparable AMP-likeness. These results demonstrate that closed-loop, reward-driven MAC provides a scalable and principled foundation for next-generation molecular design.

## 2 RELATED WORK

**AMP Design Approaches.** Traditional generative pipelines for AMP design have used adversarial or diffusion models. Early Generative Adversarial Network (GAN) based systems demonstrated feasibility for activity-based AMP generation, such as AMPGAN v2, which proposed a bidirectional

conditional GAN to steer peptide properties and showed diverse, novel sequences under conditional control (Van Oort et al., 2021). Recently, a diffusion model, Diff-AMP, unified diffusion-based generation with identification, attribute prediction, and iterative optimization in a single framework (Wang et al., 2024a). LLM-based AMP design has also been gaining traction, such as AMP Designer, a foundation model which was able to design *de novo* AMPs with broad-spectrum Gram-negative activity with a 94.4% success rate *in vitro* (Wang et al., 2025). However, most of these models rely on *ad hoc* filters or single model scores for selection, making multi-objective optimization hard to implement, and leaving a gap between evaluation outputs and trainable optimization signals that can robustly drive learning.

**LLM Multi-Agent Collaboration Systems.** LLM agents are being increasingly used in scientific discovery and evaluation. For example, the Virtual Lab employed LLM-coordinated AI agents to integrate computational protein structure prediction and modelling tools to design 92 novel SARS-CoV-2 nanobodies (Swanson et al., 2025). In materials chemistry, the OSDA (organic structure-directing agent) Agent combines an LLM with domain tools for the design of zeolite OSDAs (Hu et al., 2025). General-purpose multi-agent frameworks, such as CAMEL for role-playing-based cooperation (Li et al., 2023) and AutoGen for multi-agent dialogue and orchestration (Wu et al., 2024), have laid the groundwork for how agents collaborate and communicate to achieve common goals, demonstrating that such interactive methods can enhance model performance and outcomes. Multi-agent systems have also been explored for expert-like reviewing. For example, ReviewAgents coordinates multiple LLM reviewer roles using a structured chain-of-thought dataset to generate comments aligned with human judgments (Gao et al., 2025). However, while these multi-agent pipelines show strong coordination internally, their outputs are mostly natural language narratives, language-based sequences, or heterogeneous scores. There lacks a bridge between multi-agent consensus and executable, auditable training signals for downstream optimization. In addition, many MAC systems are open-loop, relying on human-in-the-loop orchestration and trial-and-error prompting or fine-tuning, which often yields a lot of non-executable outputs that are difficult to compile into reusable training signals.

**LLM-enhanced Reinforcement Learning and Automated Reward Design.** At the same time, there are studies exploring LLM use to guide RL. For example, RL from AI feedback (RLAIF) can replace or supplement RL from human feedback (RLHF) and actually reports performance on par with RLHF on PaLM 2 (Lee et al., 2024). Another example is Eureka, which uses coding-capable LLMs to generate and iteratively improve reward code, outperforming expert-engineered rewards on 83% of tested tasks (Ma et al., 2023). A recent survey on RL-enhanced LLMs summarizes RL's impressive performance in improving LLM capabilities (Wang et al., 2024b). However, existing approaches largely fail to integrate automated reward design with multi-agent consensus or domain-specific evidence.

## 3 PROPOSED APPROACH

### 3.1 MAC-AMP FRAMEWORK

We created MAC-AMP, a closed-loop multi-agent collaboration (MAC) system, that is designed to create novel AMPs optimized for multiple molecular properties. MAC-AMP executes the workflow through six interconnected modules, beginning with an input module and concluding with an output module. The only input required from the user is the target bacterium name and an example dataset containing AMPs and their associated minimum inhibitory concentration (MIC) values, which reflects antibacterial activity. Figure 1 outlines the entire workflow, and it is explained in detail below.

#### 3.1.1 PROPERTY PREDICTION MODULE

This module's role is to predict various AMP properties and aggregate the results into a structured record. To predict antibacterial activity, we developed a target-specific MIC predictor model. It is an LLM-based regressor adapted from BERT AmPEP60 that fine-tunes ProtBERT via transfer learning on the input dataset (8:1:1 train:validation:test split). For the remaining molecular properties, we use existing property prediction tools. AMP likelihood is predicted by Macrel 1.5 (Santos-Júnior et al., 2020), toxicity scores by ToxinPred 3.0 (Rathore et al., 2024), structural reliability scores by OmegaFold v1 (Wu et al., 2022) (EMBL-EBI AlphaFold Team, 2025), physicochemical summaries

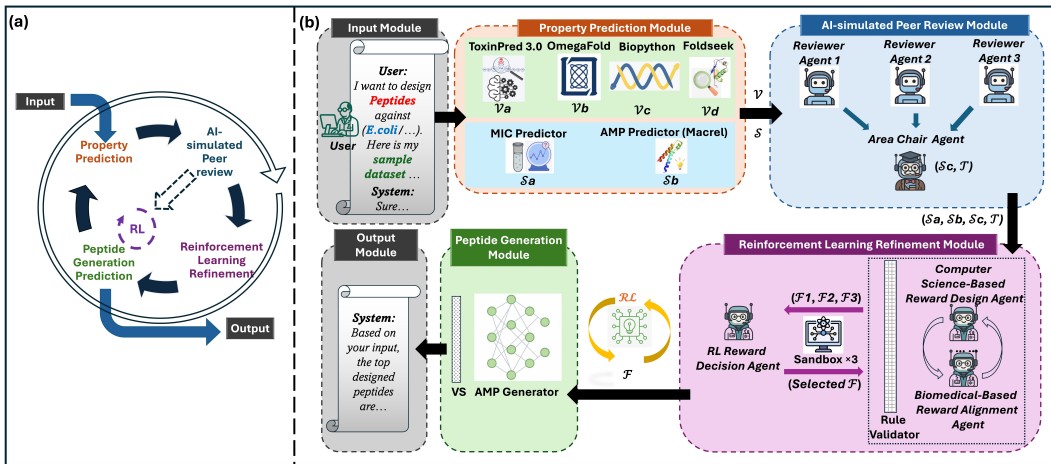

Figure 1: MAC-AMP Framework. (a) Overview of the closed-loop workflow that iteratively guides AMP design from input to output. (b) Schematic of the MAC-AMP pipeline, showing its modules and their interactions.

by ProtParam in Biopython 1.85 (Cock et al., 2009), and template similarity scores by Foldseek 10 (Van Kempen et al., 2024). Foldseek 10 selects the 10 best-performing example AMPs from the user-provided dataset (defined by the lowest MIC values) as structural templates and quantifies the similarity between generated AMP candidates and these reference peptides. Each of the property prediction tools is explained in more detail in Appendix A. Outputs of this module are partitioned into two categories: $S$, the explicit reward signal for activity; and $V$, the auxiliary evidence that constrains generation. $S$ includes two items: the antibacterial activity score provided by the target-specific MIC predictor ($S_a$), and the AMP likelihood score ($S_b$). $V$ includes four items: toxicity scores ($V_a$), structural reliability scores ($V_b$), physicochemical summaries ($V_c$), and template similarity scores ($V_d$). In the subsequent modules, different agents have different access privileges to these results. To validate the components of this module, we performed ablation studies to assess the necessity of each property prediction tool (Appendix L.1), substitution analyses to validate the ToxinPred 3.0 and MIC predictor (Appendix M.4 and M.5, respectively), and Molecular Dynamics (MD) simulations on a subset of MAC-AMP–generated AMPs to confirm using OmegaFold as a structural reliability proxy (Appendix I).

### 3.1.2 AI-SIMULATED PEER REVIEW MODULE

Motivated by committee-style deliberation in academic peer review, this module is a multi-agent system that reviews the AMPs. The overall workflow is illustrated in Figure 2 and described below.

**Reviewer Agents.** The review committee consists of three independent Reviewer agents (GPT-5, Gemini 2.5, and Perplexity), each with distinct background knowledge to ensure diverse evaluations. Inspired by multi-criteria score panels in journal peer review, each Reviewer agent evaluates candidates along four task-specific dimensions using a set of key criteria. This establishes a shared evaluation space where opinions from different Reviewer agents can be aligned and aggregated into a structured consensus. For AMP design, the four dimensions are efficiency (EFF), safety (SAFE), developmental sequence structure (DevStruct), and originality (Orig). For different tasks, these dimensions and criteria can be customized during a preparatory meeting, which is a one-on-one session where a human expert and the agent define task-specific requirements, and register these requirements as injectable knowledge (see Appendix C.3).

To make free-text reviewing quantifiable, each dimension is associated with a weighted lexicon subtable composed of Tags in the format $ID(State, Weight)$. Here, $ID$ denotes a key evaluation criterion, and $State$ is a discrete value. The determination of all Tags and their weights is designed by the Reviewer agent and then reviewed and finalized under the supervision of experienced human specialists during a preparatory meeting. The Tags are used to label Reviewer comments structurally by encoding each $State$ numerically (e.g., Low = –1, Medium = 0, High = 1). For each dimension,

a Reviewer agent first provides a free-text comment and then self-annotates by selecting up to four $IDs$ from the lexicon subtable, creating a Tag for each, and assigning a confidence score, $p$. The overall score for an AMP is computed by summing the weighted Tags scaled by the confidence score. Each Reviewer agent outputs comments, Tags, and scores for all dimensions. To ensure that the lexicon weights are decided appropriately by the agent, we performed substitution analysis to test differing lexicon weights, detailed in Appendix M.6.

**Area Chair Agent.** The Area Chair agent processes and combines the outputs of the Reviewer agents. First, it aggregates the results and drafts a meta-comment for each dimension based on the individual reviews. It then groups Tags by the same $ID$ and solves semantic conflicts (different $States$ assigned by Reviewer agents). For each dimension, it computes the mean Reviewer agent score, estimates a divergence penalty based on discrepancies in $State$, and applies this penalty to obtain a final dimension-level meta score. The module produces two outputs: a four-line delimited meta-review text ($T$) and the average meta score ($S_c$). Algorithm 1, detailed in Appendix B.1, provides the full specification of the module. To validate the components of this module, we performed ablation studies to assess the necessity of each Reviewer agent (Appendix L.2) and substitution analyses to validate the Reviewer and Area Chair agents (Appendix M.3).

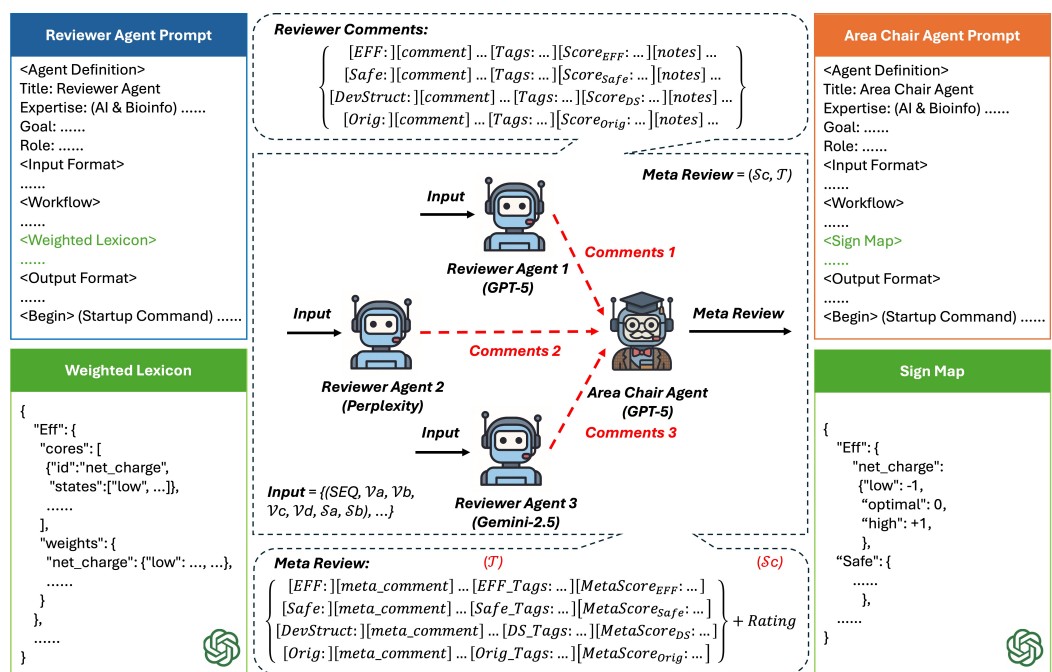

Figure 2: Overview of the Artificial Intelligence-simulated Peer Review module. Green text indicates the injectable section, while the green boxes denote content obtained from the separate preparatory meeting.

### 3.1.3 RL REFINEMENT AND PEPTIDE GENERATION MODULE

The outputs of the Property Prediction and AI-simulated Peer Review modules are fed into the RL Refinement module, which converts them into trainable optimization signals to automatically shape and adapt the reward function. The overall workflow is illustrated in Figure 3 and described in detail below.

**Candidate Reward Design.** The RL Refinement module begins by cold-starting reward design using evaluation results and consensus from a batch of example AMPs. At each stage, it reads logs from the previous stage, including the current stage index, the reward function $F$, batch-level explicit signals $S = (S_a, S_b, S_c)$, and meta-review text $T$. This information is first processed by the Computer Science (CS)-based Reward Design agent, an AI expert focused on observable signals and their mathematical properties. Guided by stage-specific prompts and constraints, this agent refines

the reward function based on the CS-relevant data (signals $S$, previous reward function $F$, stage information), intentionally ignoring the meta-review texts to maintain focus. Operating together, the Biomedical-Based Reward Alignment agent, an expert in biomedical and AMP design, analyzes the meta-review texts and integrates domain knowledge to propose alignment-oriented revisions to the CS-based Reward Design agent's candidate reward functions. This agent accesses only the meta-review texts, avoiding distraction by non-biomedical signals. Candidate rewards are then filtered by a rule-based validator for executability and constraint compliance, yielding a candidate set of three reward functions.

Next, the module clones the Peptide Generation module into a sandbox, runs short simulated training for each candidate reward function, and the RL Reward Decision agent selects the option with the best overall performance via Pareto optimization. The chosen reward function and sandbox logs are returned to the CS–Biomedical Reward Design agent team for further refinement. This inner loop iterates three times.

To validate the components of this module, we performed ablation studies to assess the necessity of each component, detailed in Appendix L.3.

**Peptide Generation Module.** After the inner loop, the selected reward function is compiled into a Proximal Policy Optimization (PPO) objective, which guides the Peptide Generation module through AMP design. The generator, inherited from the AMP-Designer architecture (Wang et al., 2025), is a GPT-2 auto-regressive model with a trainable soft prompt that injects domain prior knowledge. Generated AMPs are evaluated using the Property Prediction and AI-simulated Peer Review modules, producing batch-level meta-review texts and scores. These evaluations are incorporated into a reward-based PPO strategy to calculate the loss and update generator parameters every epoch.

**Stage-Based Adaptive Optimization.** A stage is defined as 15 epochs under the same reward function and PPO strategy. At the end of each stage, the RL module aggregates all evaluation logs and adaptively redesigns the reward, yielding an optimized PPO strategy for the next stage. This adaptive redesign is repeated three times, allowing the reward function to co-evolve with real-time feedback and multi-agent consensus. Algorithm 2, detailed in Appendix B.2, provides the full specification of the module. Although the number of epochs defining a stage can be adjusted, we performed substitution analyses to evaluate the effects of increasing or decreasing the default (15 epochs), detailed in Appendix M.1.

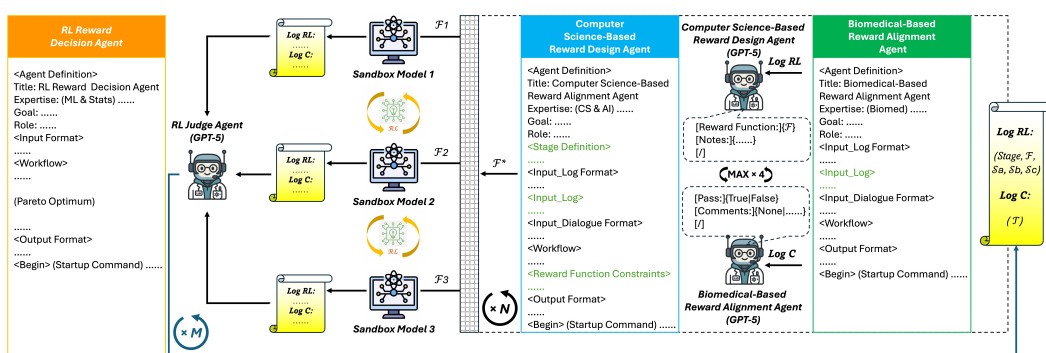

Figure 3: Overview of the Reinforcement Learning Refinement module.

**Communication Logs.** Through every batch and training stage, communication among modules primarily relies on reading and writing structured system logs. Each log contains six fields: Stage Number (0, 1, 2, or $IN$, where $IN$ denotes inner-loop sandbox logs), reward function ($F$), $S_a$, $S_b$, $S_c$ and $T$. This structured logging ensures end-to-end auditability.

## 3.2 PROXIMAL POLICY OPTIMIZATION

In MAC-AMP, a PPO strategy is instantiated from the stage-selected reward and applied to update the generation module in a sequence-level decision setting. Conditioned on a trainable soft prompt and a partial prefix, the policy $\pi_\theta$ auto-regressively generates a complete sequence $\theta = (a_1, \dots, a_T)$.

The value network produces token-wise estimates, which are averaged across positions to yield the sequence baseline $\bar{V}_\phi$.

The standardized advantage is:

$$A = norm(R - \bar{V}_\phi) \tag{1}$$

where $R$ is the sequence-level reward for a sampled trajectory, $\bar{V}_\phi$ is the sequence baseline obtained by averaging the token-wise value predictions, and $norm(\cdot)$ denotes batch-wise standardization to zero mean and unit variance.

The clipped surrogate loss is:

$$L_{policy}(\theta) = \mathbb{E}[min(r(\theta)A, clip(r(\theta), 1 - \epsilon, 1 + \epsilon)A)] \tag{2}$$

where $r(\theta)$ is the probability ratio between the current policy and the reference policy for the sampled action, $\epsilon > 0$ is the PPO clipping hyperparameter, $clip(r(\theta), 1 - \epsilon, 1 + \epsilon)$ clamps this ratio to the interval $[1 - \epsilon, 1 + \epsilon]$, and $\mathbb{E}[\cdot]$ denotes expectation over sampled trajectories and time steps.

The value regression term is:

$$L_{value}(\phi) = \mathbb{E}[(\bar{V}_\phi - R)^2] \tag{3}$$

where $L_{\text{value}}(\phi)$ trains the value network by minimizing the mean-squared error between the predicted baseline $\bar{V}_\phi$ and the reward $R$.

The entropy regularization term is:

$$L_{ent}(\theta) = \mathbb{E}_{t \epsilon gen}[H(\pi_\theta(\cdot|s_t))] \tag{4}$$

where $s_t$ is the decoder state at generation step $t$, $\pi_\theta(\cdot \mid s_t)$ is the policy distribution over next tokens at that step, $H(\cdot)$ is the Shannon entropy of this distribution, and $\mathbb{E}_{t \in \text{gen}}[\cdot]$ denotes averaging over the generation time steps $t \in$ gen.

The total loss is:

$$L = L_{policy} + c_v L_{value} - c_e L_{ent} \tag{5}$$

where $c_v > 0$ and $c_e > 0$ are scalar hyperparameters that weight the value regression and entropy regularization terms, respectively. $L$ is the overall training objective minimized during policy updates.

The training is conducted in rounds. A batch of sequences is sampled, $R$, $\bar{V}_\phi$, and $A$ are computed, and a single gradient update is performed on that batch.

In addition, a schema-driven prompting framework is incorporated to enable multi-agent collaboration that is generalizable, transferable, and reusable across a wide range of applications and is inspired by human organizational practices. The structural design of this collaboration framework is described in Appendix C.

To assess whether stage-wise RL in MAC-AMP remains stable during training and to verify that the Reviewer agents do not induce feedback collapse, overfitting to internal Reviewer agent biases, or reward hacking, additional analyses were conducted and are detailed in Appendix D.

## 4 EXPERIMENTS

### 4.1 AMP GENERATION TESTING

**Bacterial Targets.** AMP design performance was evaluated against five bacterial targets. First, *Escherichia coli (E. coli)*, which has been associated with multiple infections and diseases (e.g., urinary tract infections (Totsika et al., 2012), neonatal meningitis (Bonacorsi & Bingen, 2005)). Four other ESKAPE pathogenic bacterial strains were also analyzed: *Staphylococcus aureus (S. aureus)*, *Pseudomonas aeruginosa (P. aeruginosa)*, *Klebsiella pneumoniae (K. pneumoniae)*, and *Enterococcus faecium (E. faecium)*. ESKAPE pathogens are a group of bacteria that the World Health Organization (WHO) and U.S. Centers for Disease Control and Prevention (CDC) flag as major threats because they are leading causes of hospital-acquired infections and often display AMR (Miller & Arias, 2024). Together, these targets also reflect diverse Gram staining profiles, as *S. aureus* and *E. faecium* are Gram-positive, while the remaining are Gram-negative.

**Dataset Preparation & Pre-Processing**. For each bacterial target, AMP sequences were collected along with their corresponding MIC values. MIC represents the lowest concentration of a compound, or in this case, AMP, that inhibits bacterial growth. This data was sourced from two public databases: DBAASP v3 (Pirtskhalava et al., 2021) and dbAMP 3.0 (Yao et al., 2025). AMP sequences were standardized by converting to uppercase, removing whitespace, and retaining only canonical IUPAC letters, and entries with non-standard residues were excluded. Duplicate sequences were removed, and replicate MIC measurements for the same sequence were aggregated using the geometric mean to obtain a single value. AMP sequences were represented in IUPAC single-letter codes, and their MIC values ($\mu$g/mL) were log10-transformed to serve as labels. The final datasets contained 3,818 AMP examples for *E. coli*, 2,644 for *S. aureus*, 2,458 for *P. aeruginosa*, 838 for *K. pneumoniae*, and 352 for *E. faecium*.

**Target-Specific AMP Testing.** *E. coli*, *S. aureus*, and *P. aeruginosa* were used to test target-specific design tasks. In each test, one of the bacterial target datasets was provided by the user and passed via the input module. During generation, the generation head produced 1,000 AMP candidates, of which only the top 30 (ranked by predicted MIC) were retained for downstream performance analysis. This procedure was repeated three times, resulting in a total of 90 AMPs.

**Broad-Spectrum Activity Testing.** To test the broad-spectrum activity of the generated AMPs, a separate MIC predictor was trained for each bacterial strain, which was then used to evaluate the *E. coli*–designed AMPs and assessed their generalization across species. Further analyses, including evaluation with an external MIC predictor (APEX 1.1) and motif analysis of broad-spectrum AMPs, are detailed in Appendix F.

**Baseline Models.** MAC-AMP was compared against two categories of generative baselines: LLM-based and non-LLM traditional, as well as a real-world AMP dataset. The two LLM-based baselines are AMP Designer and BroadAMP GPT. AMP-Designer is a comprehensive framework for AMP design that integrates GPT, prompt tuning, contrastive learning, knowledge distillation, and RL (Wang et al., 2025). BroadAMP GPT employs transformer-based generation and deep learning-guided screening for AMP design (Li et al., 2025). The two non-LLM-traditional baselines are PepGAN and Diff-AMP. PepGAN is a GAN-based model based on LeakGAN, a state-of-the-art sequence generator, but incorporates an activity predictor that is trained separately with positive and negative examples together (Tucs et al., 2020). Diff-AMP is a diffusion-based model that, alongside diffusion, employs pre-training and iterative optimization technologies to advance AMP design (Wang et al., 2024a). Details on how baseline model testing was performed, and the real-world AMP dataset was chosen, can be found in Appendix E.

## 4.2 ENVIRONMENT DETAILS AND COMPUTATIONAL COSTS

Experiments were run in PyTorch on NVIDIA A100 GPUs. The AMP generator is a GPT-2 small (12 layers, 12 heads, hidden size 768) augmented with a 10-token soft prompt, using a BERT-style amino-acid tokenizer. During PPO, the policy is optimized and a GPT-2 value head with Adam (lr = 5e-5) and gradient-norm clipping at 1.0. The Peptide Generation module uses top-k = 50 / top-p = 0.95 with temperature = 1.0.

To train MAC-AMP for AMP prediction on this environment, it took 47.61 GPU hours, 853 API calls, 9106 MB of peak memory, and incurred a total API token cost of $36.56 USD.

## 4.3 RESULTS

**Target-Specific AMP Testing.** Across the three single-task bacterial targets (*E. coli, S. aureus, and P. aeruginosa*), MAC-AMP consistently achieves the best antibacterial activity, toxicity, and structural reliability scores, as shown in Table 1. This proves that MAC-AMP enhances target-specific efficacy, while imposing effective safety constraints that suppress potential toxicity and implement effective assessments of structural and physicochemical properties that are leveraged to steer AMP generation toward more stably foldable sequences with fewer structural hallucinations. Although BroadAMP-GPT shows a slight advantage on AMP likelihood, it performs notably worse on toxicity and structural reliability. This indicates that, without heavily sacrificing AMP discriminability, our approach allocates optimization capacity to multiple objectives, ultimately delivering performance that better aligns with real-world research and development needs.

Table 1: Property scores of antimicrobial peptides (AMP) generated by MAC-AMP compared to baseline models and real-world AMP datasets across bacterial species

| Model | Antibacterial Activity (↑) | AMP Likelihood (↑) | Toxicity (↓) | Structural Reliability (↑) |
|---|---|---|---|---|
| *Escherichia coli (E. coli)* | | | | |
| MAC-AMP | **0.943 ± 0.008** | 0.797 ± 0.012 | **0.154 ± 0.008** | **0.873 ± 0.009** |
| AMP-Designer | 0.807 ± 0.021 | 0.811 ± 0.011 | 0.251 ± 0.024 | 0.817 ± 0.017 |
| BroadAMP-GPT | 0.831 ± 0.025 | **0.821 ± 0.018** | 0.246 ± 0.033 | 0.763 ± 0.023 |
| PepGAN | 0.823 ± 0.023 | 0.572 ± 0.035 | 0.247 ± 0.064 | 0.637 ± 0.026 |
| Diff-AMP | 0.822 ± 0.006 | 0.554 ± 0.036 | 0.235 ± 0.072 | 0.752 ± 0.020 |
| Real-World- top K | 0.894 ± 0.014 | 0.807 ± 0.030 | 0.558 ± 0.068 | 0.846 ± 0.022 |
| *Staphylococcus aureus (S. aureus)* | | | | |
| MAC-AMP | **0.931 ± 0.007** | 0.849 ± 0.008 | **0.137 ± 0.011** | **0.837 ± 0.009** |
| AMP-Designer | 0.809 ± 0.023 | 0.807 ± 0.012 | 0.225 ± 0.022 | 0.801 ± 0.017 |
| BroadAMP-GPT | 0.823 ± 0.025 | **0.858 ± 0.014** | 0.448 ± 0.062 | 0.763 ± 0.025 |
| PepGAN | 0.901 ± 0.019 | 0.742 ± 0.014 | 0.231 ± 0.059 | 0.644 ± 0.021 |
| Diff-AMP | 0.926 ± 0.013 | 0.535 ± 0.023 | 0.281 ± 0.130 | 0.764 ± 0.023 |
| Real-World- top K | 0.746 ± 0.033 | 0.742 ± 0.031 | 0.543 ± 0.070 | 0.769 ± 0.023 |
| *Pseudomonas aeruginosa (P. aeruginosa)* | | | | |
| MAC-AMP | **0.917 ± 0.008** | 0.851 ± 0.006 | **0.110 ± 0.014** | **0.850 ± 0.010** |
| AMP-Designer | 0.839 ± 0.018 | 0.816 ± 0.011 | 0.243 ± 0.024 | 0.799 ± 0.019 |
| BroadAMP-GPT | 0.842 ± 0.031 | **0.858 ± 0.014** | 0.449 ± 0.067 | 0.772 ± 0.022 |
| PepGAN | 0.912 ± 0.013 | 0.747 ± 0.013 | 0.248 ± 0.056 | 0.664 ± 0.027 |
| Diff-AMP | 0.907 ± 0.006 | 0.594 ± 0.022 | 0.201 ± 0.072 | 0.766 ± 0.021 |
| Real-World- top K | 0.802 ± 0.023 | 0.785 ± 0.031 | 0.568 ± 0.064 | 0.820 ± 0.024 |

**Broad-Spectrum Activity Testing.** When evaluating the broad-spectrum potential of the anti-*E. coli* AMPs, overall, MAC-AMP peptides showed the strongest generalization, achieving the highest antibacterial activity scores for more non–*E. coli* species than any other model (Table 2). *E. coli*-specific AMPs show excellent generalization in other Gram-negative species (*P. aeruginosa, K. pneumoniae*), indicating that the learned physicochemical patterns (e.g., cationic charge density, hydrophobicity, appropriate length) transfer well across Gram-negative bacterial strains, likely due to their shared outer membrane architecture. Interestingly, we also observe strong generalization to *E. faecium*, a Gram-positive species, while *S. aureus* shows somewhat reduced activity. This suggests that while cell envelope structure influences AMP susceptibility, our results demonstrate that effective generalization is achievable across both Gram-negative and Gram-positive species. The strong activity against *E. faecium* shows that Gram-positive classification doesn't preclude broad-spectrum efficacy. Species-specific factors may modulate activity levels (as seen with *S. aureus*), but this simply indicates that some additional optimization or validation may be beneficial for certain targets.

Table 2: Antibacterial activity scores of *Escherichia coli*-targeted antimicrobial peptides against other bacterial strains

| Model | *E. coli* | *S. aureus* | *P. aeruginosa* | *K. pneumoniae* | *E. faecium* |
|---|---|---|---|---|---|
| MAC-AMP | 0.94 ± 0.01 | 0.81 ± 0.03 | 0.94 ± 0.00 | 0.98 ± 0.00 | 0.95 ± 0.01 |
| AMP-Designer | 0.81 ± 0.02 | 0.81 ± 0.02 | 0.85 ± 0.01 | 0.96 ± 0.01 | 0.96 ± 0.01 |
| BroadAMP-GPT | 0.83 ± 0.02 | 0.82 ± 0.03 | 0.87 ± 0.02 | 0.96 ± 0.01 | 0.97 ± 0.01 |
| PepGAN | 0.82 ± 0.02 | 0.89 ± 0.02 | 0.91 ± 0.01 | 0.98 ± 0.00 | 0.96 ± 0.01 |
| Diff-AMP | 0.82 ± 0.01 | 0.91 ± 0.01 | 0.94 ± 0.01 | 0.98 ± 0.00 | 0.93 ± 0.01 |

**Comparison of MAC-AMP and Real-World AMPs.** In addition, using Uniform Manifold Approximation and Projection (UMAP), MAC-AMP candidate peptides for *E. coli* were projected alongside the *E. coli* AMP training dataset (real-world AMPs) and a UniProt background set con-

sisting of peptide sequences extracted from UniProtKB reference proteomes, used to represent the broader space of naturally occurring peptides (Consortium, 2025). Figure 4a shows that MAC-AMP candidates cluster within or along the edges of the blue density wells defined by real-world AMPs, rather than being scattered across the broader UniProt space. This spatial alignment indicates that MAC-AMP has learned AMP-like features while also populating multiple subregions of the "AMP manifold", balancing fidelity with exploration of underrepresented neighbourhoods that may yield novel activity. Figure 4b and Figure 4c illustrate two MAC-AMP-generated AMPs for *E. coli*, both of which display canonical AMP chemistries (e.g., Lys/Arg enrichment, aromatic/hydrophobic residues such as Trp, Leu, Ile, and Val) consistent with electrostatic membrane association and amphipathic disruption. This supports our finding that MAC-AMP concentrates sampling in biophysically plausible, *E. coli*–relevant regions of sequence space while maintaining chemotype diversity. To further assess similarity, 2000 MAC-AMP-generated *E. coli* AMPs were compared against baseline models and real-world AMPs in Appendix K.

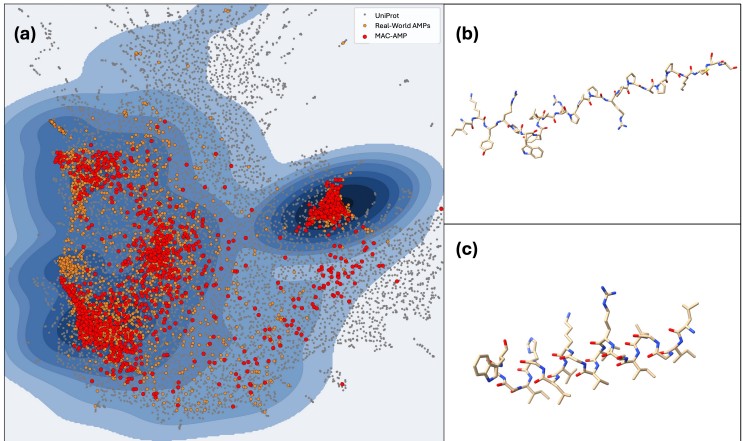

Figure 4: (a) UMAP of the peptide chemical space for *Escherichia coli (E. coli)* inhibition. Gray dots show UniProt peptides, orange dots show real-world *E. coli* AMPs, and red dots are MAC-AMP candidate AMPs. Nested blue contours indicate increasing kernel density from these AMPs. (b,c) Sequence diagrams of two example MAC-AMP designed *E. coli* AMPs.

The 90 generated anti-*E. coli* AMPs were evaluated for novelty, antibacterial motifs, and structural stability in Appendices G, H, and I, respectively. Also, The sequences and biophysical properties of six MAC-AMP-generated *E. coli* AMPs are analyzed in Appendix J. Finally, to evaluate MAC-AMP's cross-domain transferability, we tested it on an English table-to-text generation task, where the model generates a one-sentence description from a table and highlighted cells. The details and results are detailed in Appendix N.

## 5 CONCLUSION

In conclusion, MAC-AMP is the first fully autonomous, closed-loop multi-agent system for AMP design, reframing AMP generation as a coordinated multi-agent optimization problem. By translating structured consensus on activity, safety, stability, and novelty into executable reward signals, it enables stable, multi-objective optimization with full auditability. MAC-AMP surpasses existing models in antibacterial potency, toxicity, and structural reliability while maintaining AMP-likeness, exploring underrepresented yet biophysically plausible regions of sequence space. More broadly, it provides a scalable, interpretable framework for next-generation molecular design. Limitations and future directions are discussed in Appendix O. All code and data can be found at `https://github.com/CLMFAP/MAC-AMP_v1/`.

## 6 ACKNOWLEDGMENTS

This work was supported in part by the Canada Research Chairs Tier II Program (CRC-2021-00482), the Canadian Institutes of Health Research (PLL 185683, PJT 190272, PJT204042, CFA - 205059), the Natural Sciences, Engineering Research Council of Canada (RGPIN-2021-04072,ALLRP 602759-24) and The Canada Foundation for Innovation (CFI) John R. Evans Leaders Fund (JELF) program (#43481).

## 7 ETHICS STATEMENT

This paper presents MAC-AMP, a novel model for the design and generation of AMPs. While its primary focus is accelerating AMP discovery, MAC-AMP could also be applied in broader biomedical research and be adapted for other domains. At present, we do not anticipate any direct or immediate ethical concerns associated with the development or use of this model. In particular, we have considered the potential societal impacts of MAC-AMP and do not believe there are any concerns at this time. MAC-AMP users must apply the model responsibly, ensure transparency regarding its capabilities and limitations, and verify that any outputs are used safely and ethically. Finally, we emphasize that MAC-AMP is intended as a research tool to augment human expertise, not to replace critical scientific judgment. We encourage continued evaluation of potential risks and responsible deployment.

## 8 REPRODUCIBILITY STATEMENT

We have made multiple efforts to ensure reproducibility throughout our study. We have included extensive details in the main manuscript and additional sections in our Appendix that provide algorithms and more in-depth explanations of the theoretical concepts.

The complete description of the data processing steps and sources of data is detailed in Section 4.1. The exact tools used in our Property Prediction module are detailed in Appendix A. The algorithms for our AI-simulated Peer Review and RL Refinements modules are detailed in Appendix B. The theory behind our proximal policy optimization and structural design of MAC is detailed in Section 3.2 and Appendix C, respectively. We have included the exact details of our baseline model training steps in Appendix E. We have also released all code and data at `https://github.com/CLMFAP/MAC-AMP_v1/`. By providing source code, comprehensive details on the methodology, algorithms, training environment, hyperparameters, and data processing, we commit to fully transparent and reproducible research.

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

APPENDIX - MAC-AMP: A CLOSED-LOOP MULTI-AGENT COLLABORATION SYSTEM FOR MULTI-OBJECTIVE ANTIMICROBIAL PEPTIDE DESIGN

APPENDIX TABLE OF CONTENTS

# A  PROPERTY PREDICTION MODULE TOOLS

**ToxinPred 3.0.** ToxinPred 3.0 is a sequence-based general toxicity predictor for peptides that outputs a probability between 0 and 1, where higher values indicate higher toxicity risk (Rathore et al., 2024).

**OmegaFold v1.** OmegaFold is a computational tool that predicts the 3D structure of a protein directly from its amino acid sequence. Along with the predicted structure, it provides a pLDDT score for each residue, which ranges from 0 to 1, as a structural reliability indicator, where higher values indicate higher confidence (Wu et al., 2022; EMBL-EBI AlphaFold Team, 2025).

**Biopython 1.85.** Biopython contains a collection of Python tools for bioinformatics, from which the ProtParam module is used for physicochemical profiling. This module outputs a concise text summary of sequence properties such as length, molecular weight, theoretical isoelectric point, and Grand Average of Hydropathy (GRAVY) (Cock et al., 2009).

**Foldseek 10.** Foldseek 10 is a structure and sequence alignment tool that computes similarity between protein structure sets (including peptide sequences), and reports a normalized similarity score between 0 and 1, where a higher score indicates greater similarity (Van Kempen et al., 2024).

**Macrel 1.5.** Macrel 1.5 is a sequence-based AMP classifier that outputs an AMP likelihood score between 0 and 1, where higher values indicate a higher predicted probability of being an AMP (Santos-Júnior et al., 2020).

**Target -Specific MIC Predictor.** We designed an LLM-based regressor, adapted from BERT Am-PEP60 (Cai et al., 2025), that fine-tunes ProtBERT via transfer learning (Elnaggar et al., 2022) to predict target-specific MIC values for AMP sequences. In our system, it is automatically trained for each target using the user-provided, real-world AMP dataset. The raw output is the MIC ($\mu$g/mL), representing the minimum concentration that inhibits visible growth (Kowalska-Krochmal & Dudek-Wicher, 2021). To make it compatible with a higher-is-better RL reward, the MIC is transformed using a sigmoid-shaped function into an antibacterial activity score between 0 and 1, inclusive. Higher scores correspond to lower MIC values, which reflect stronger target-specific activity.

# B  ALGORITHMS

## B.1  AI-SIMULATED PEER REVIEW MODULE ALGORITHM

---

**Algorithm 1** AI-Simulated Peer Review module Algorithm

---

1: **function** REVIEWMODULE($batch$, $lexicons$, $signmap$)
2:     **for** $r \in \{R1, R2, R3\}$ **do**
3:         $R[r] \leftarrow$ RUNREVIEWER($r, batch, lexicons$)
4:     **end for**
5:     $AC \leftarrow$ AGGREGATEBYAC($R, signmap$)
6:     $S \leftarrow$ COMPUTESCORES($R, AC$)
7:     **return** $AC, S$
8: **end function**

9: **function** RUNREVIEWER($r$, $batch$, $lexicons$)
10:     **for** $a \in \{Eff, Safe, DevStruct, Orig\}$ **do**
11:         $comment[a] \leftarrow$ generate $\leq 1500$ chars based on analysis of $(S, V)$
12:         $tags[a] \leftarrow$ select $\leq 4$ (id, state, p) via $lexicons$; $p \in \{1.00, 0.85, 0.60, 0.40\}$
13:         $score[a] \leftarrow \sum w(id, state) \cdot p$ over $tags[a]$
14:     **end for**
15:     **return** $\{comment, tags, score\}$
16: **end function**

17: **function** AGGREGATEBYAC($R$, $signmap$)
18:     **for** $a \in \{Eff, Safe, DevStruct, Orig\}$ **do**
19:         $meta[a] \leftarrow$ concise summary of agreements based on all $comment[a]$
20:         $G \leftarrow$ group all reviewers' tags by id
21:         $Dist[a] \leftarrow$ sorted signs from $signmap$ for ids with $\geq 2$ hits; keep all-zero
22:         $Num[a] \leftarrow |Dist[a]|$               ▷ number of ids with overlapped states
23:     **end for**
24:     **return** $\{meta, G, Dist, Num\}$
25: **end function**

26: **function** COMPUTESCORES($R$, $AC$)
27:     **for** $a \in \{Eff, Safe, DevStruct, Orig\}$ **do**
28:         $\bar{S}[a] \leftarrow$ mean($score[a]$ over reviewers)
29:         $d\_list \leftarrow [\,]$
30:         **for** each $(id, S)$ in $Dist[a]$ **do**       ▷ $S$ is tuple of signs, e.g. $(-1, -1, 1)$ or $(0, 0)$
31:             **if** $\max(|S|) = 0$ **then**                   ▷ all-zero special case
32:                 $d\_id \leftarrow 0$
33:             **else**
34:                 $d\_id \leftarrow 1 - |\text{mean}(S)|$
35:             **end if**
36:             **if** $d\_id = 0$ **then**
37:                 **continue**
38:             **end if**
39:             append($d\_list, d\_id$)
40:         **end for**
41:         $D[a] \leftarrow$ mean($d\_list$) if $d\_list \neq \emptyset$ else 0
42:         $\gamma[a] \leftarrow \text{clip}_{[0.6, 1.0]}(1 - 0.6 \cdot D[a])$
43:         $\text{meta}[a] \leftarrow \gamma[a] \cdot \bar{S}[a]$
44:     **end for**
45:     $overall \leftarrow$ mean($\text{meta}[a]$ over aspects)
46:     **return** $\{\bar{S}, D, \gamma, \text{meta}\}, overall$
47: **end function**

---

## B.2 REINFORCEMENT LEARNING REFINEMENT MODULE ALGORITHM

---

**Algorithm 2** Reinforcement Learning Refinement module Algorithm

---

1: candidates ← 3                                                  ▷ number of candidate reward functions per inner loop
2: inner_rounds ← 3                                                ▷ sandbox optimize rounds
3: dialog_max ← 4                                                  ▷ agent ↔ critical review turns
4: u_sandbox ← 5                                                   ▷ sandbox updates per candidate
5: u_outer ← 15                                                    ▷ outer training updates per stage
6: stages ← {1, 2, 3}                                             ▷ exploration → balance → convergence

7: **function** RLMODULE(gen_model, reviewer_module)
8:     out ← $init\_cold\_start(reviewer\_module)$
9:     **for** $s \in stages$ **do**
10:        f* ← $run\_inner\_loop(gen\_model, out, s)$
11:        out ← $run\_outer\_train(gen\_model, f*, s, reviewer\_module, out)$
12:    **end for**
13:    **return** out
14: **end function**

15: **function** RUN_INNER_LOOP(gen_model, out, stage_id)
16:    snap ← $snapshot(gen\_model)$
17:    **for** $r \in \{1..inner\_rounds\}$ **do**
18:        c ← []
19:        **for** $k \in \{1..candidates\}$ **do**
20:            f ← $co\_design(out, stage\_id)$
21:            **if** $f \neq None$ **then**
22:                c.append(f)
23:            **end if**
24:        **end for**
25:        logs ← $run\_sandbox(snap, c, u\_sandbox)$
26:        m ← $rl\_decision\_select(logs)$
27:        f_best, l_best ← $c[m], logs[m]$
28:        feedback_to_agents(f_best, l_best)
29:        snap ← $restore(snap)$
30:    **end for**
31:    **return** f_best
32: **end function**

33: **function** CO_DESIGN(out, stage_id)
34:    p ← $agent\_propose(out, stage\_id)$
35:    **for** $t \in \{1..dialog\_max\}$ **do**
36:        pass, cmts ← $critical\_review(p, out, stage\_id)$
37:        **if** pass = true **then**
38:            **break**
39:        **end if**
40:        p ← $agent\_revise(p, cmts)$
41:    **end for**
42:    **if** $\neg rule\_validate(p)$ **then**
43:        **return** None
44:    **end if**
45:    **return** p
46: **end function**

---

```
47: function RUN_SANDBOX(snap, cand_set, u)
48:     logs ← {}
49:     for each f ∈ cand_set do
50:         m ← clone(snap)
51:         for u_i ∈ {1..u} do
52:             r ← train_step(m, f)
53:             t ← reviewer_module_eval(m)
54:             append_inside_log(logs[f], stage="in", f, r, t)
55:         end for
56:     end for
57:     return logs
58: end function

59: function RL_DECISION_SELECT(logs)
60:     winner ← argmax_by_rules(logs)
61:     return winner
62: end function

63: function RUN_OUTER_TRAIN(gen_model, f, s, reviewer_module, out)
64:     for u_i ∈ {1..u_outer} do
65:         r ← train_step(gen_model, f)
66:         t ← reviewer_module_eval(gen_model)
67:         write_outside_logs(out, stage=s, f, r, t)
68:     end for
69:     return out
70: end function

71: function WRITE_OUTSIDE_LOGS(out, stage, f, r, t)
72:     append(out.agent, record(stage, f, r.sa, r.sb, r.sc))
73:     append(out.critical, record(stage, t))
74: end function
```

## C   DETAILS REGARDING STRUCTURAL DESIGN OF MULTI-AGENT COLLABORATION

This section outlines the overarching design principles that govern all agent-based modules and the agents within them. Adherence to these structured rules ensures that each agent consistently fulfills its designated role and that inter-agent communication and collaboration remain accurate, stable, and effective.

Inspired by human organizational practices, the MAC framework formalizes roles, operating procedures, and human onboarding practices into a unified structure, enabling agents to coordinate reliably without altering team composition or core methodology. It comprises three components: (1) a role-based agent profile (representing role establishment) that anchors identity and responsibility; (2) a role-bound operating contract (representing operating manual) specifying standardized input/output formats, workflow steps, startup commands, and reserved slots for additional information; and (3) an knowledge injection (representing human onboarding practice) via preparatory-meeting decisions. When pivoting to a new task, only the injectable content is updated while profiles and contracts remain unchanged, and inter-agent communication occurs via role-dependent access to local and global logs, ensuring structured, stable collaboration.

### C.1   ROLE-BASED AGENT PROFILE

We define the role-based agent profile as the long-lived professional identity of a single agent that specifies who the agent is, what competencies it commands, what outcomes it pursues, and where its responsibility boundaries lie. The profile remains invariant across tasks and domains, enabling the accumulation of role experience. For this part, we adopt the four-anchor specification: Title,

Expertise, Goal, and Role as introduced by the AI Virtual Lab (Swanson et al., 2025), explained below:

***Title:*** A concise, professional, and task agnostic position name that precisely denotes the agent's occupational identity.

***Expertise:*** A brief description of the core disciplinary knowledge and methodological capabilities the position relies on, emphasizing a stable competence scope and toolbox.

***Goal:*** A results-oriented statement of the agent's success criteria and optimization target while avoiding overlap with other positions.

***Role:*** A clear charter of responsibility boundaries and decision authority within the collaboration pipeline, together with the obligations and principles that govern interaction with other positions.

## C.2    SCHEMA-DRIVEN SYSTEM PROMPTS

For a single agent, the core elements consist of four parts: Base Model, Parameter Settings, User Prompt, and System Prompt. Base Model refers to the API service on which the agent relies and determines differences in its default knowledge background and working style. Parameter Settings are used to adjust the agent's behaviour. User Prompt exists as dialogue and is typically used for intra-team communications in the framework. Inter-team communications are performed by reading logs via injecting local or global log entries into the corresponding section in the System Prompt of the agent under access control. System Prompt is the primary design target and implements the three-part abstraction: role-based agent profile, role-bound operating contract, and injectable section. We adopt a schema-driven approach to modularize the System Prompt and refine it into the following sections:

***Agent Definition:*** Insert the agent's role-based agent profile here.

***Input Format:*** The first part of the role-bound operating contract, declaring the input structures that the agent may receive. When multiple inputs are required simultaneously, list different input types on separate lines. Common inputs include dialogue history from the User Prompt and log entries injected into the System Prompt.

***Workflow:*** The second part of the role-bound operating contract, providing a role-based general workflow guideline.

***Output Format:*** The third part of the role-bound operating contract, declaring the agent's output format so that free-text responses are constrained to a fixed structure and can be written to logs.

***Startup Command:*** The standard command that requests the agent to start working.

***Injectable Section:*** Used for task-specific knowledge customization and injection as a functional unit. When needed, inter-team meeting logs or other multi-source inputs can also be injected here.

## C.3    KNOWLEDGE INJECTION VIA PREPARATORY MEETINGS

We introduce a general, transferable knowledge-injection design that treats task intent as an interchangeable specification, decoupling domain guidance from the core role to enable rapid cross-task adaptation while preserving reproducibility. The two main parts of the knowledge-injection design are described below:

***Preparatory Meeting:*** For a task in a new domain, a preparatory meeting is held between a human expert and the corresponding agent before deployment. During the meeting, the agent retains only its role-based profile, with the role-bound operating contract temporarily left empty. The human expert describes the new task's specific process requirements and, together with the agent, drafts a detailed plan as the meeting output.

***Knowledge Injection:*** The finalized content from the preparatory meeting is written into the agent's System Prompt under the Injectable Section as the task-specific knowledge and preferences required for subsequent execution. The role-based profile and the role-bound operating contract remain unchanged.

## D    EMPIRICAL ANALYSIS OF REWARD VARIANCE AND CLOSED-LOOP STABILITY

To assess whether stage-wise RL in MAC-AMP remains stable throughout training and to verify that the multi-agent evaluators do not induce feedback collapse, overfitting to internal Reviewer agent biases, or reward hacking, the reward–episode trajectories were analyzed across all three RL stages. This analysis tracked the total reward as well as its components ($S_a$, antibacterial activity score; $S_b$, AMP likelihood score; and $S_c$, average meta score) over training episodes and examined how each term evolved relative to the others. In addition, the variance band of the total reward was evaluated over time and compared against the component-wise trajectories. This allowed detection of potential reward-hacking signatures, such as one sub-term being pushed to an extreme while others degrade, as well as signs of Reviewer agent-signal collapse or instability in the closed-loop feedback.

In Figure A1, the trajectories of $S_a$, $S_b$, and $S_c$ display the intended stage-specific behaviour. In the early stage, after an initial drop when the reward is reset, the policy primarily increases $S_b$, reflecting an exploration-oriented focus on AMP-likeness. In the mid stage, once $S_a$ and $S_b$ have stabilized at reasonable levels, the system increases $S_c$, which aggregates multi-agent assessments involving toxicity, structural reliability, and other constraints. In the final stage, after constraint-related signals plateau, optimization shifts toward further elevating $S_a$ to enhance predicted antibacterial activity.

Across all stages, the total reward rises without any indication of reward hacking. There are no trajectories in which total reward increases by sharply degrading one component while over-optimizing another. Instead, the three reward components co-evolve in a manner consistent with the intended stage-wise objectives. Likewise, the variance band of the total reward does not collapse, indicating that the stage-wise reward structure acts as an implicit regularizer that prevents any single objective from being disproportionately exploited within the current three-stage training horizon.

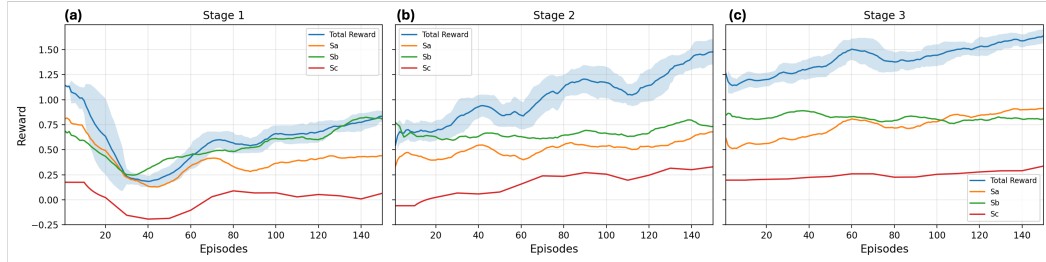

Figure A1: Three-stage reward–episode learning curves for MAC-AMP. The curves illustrate how the total reward and its components ($S_a$, antibacterial activity score; $S_b$, AMP likelihood score; and $S_c$, average meta score) evolve during (a) the early stage, (b) the middle stage, and (c) the late stage. The light blue area indicates the moving variance of the total reward.

## E    BASELINE MODELS TRAINING

For all generative baselines, a unified protocol is used to generate AMPs for each target organism: each model independently generates 1,000 candidate AMPs, evaluates them with the same target-specific MIC predictor to obtain an antibacterial activity prediction score, where a higher score indicates a lower predicted MIC value, and then selects the top-k (k=30) from that batch as the representative set for the run. Each model is run three times with different random seeds, yielding 3×30 representative sequences for evaluation and statistical analysis. De-duplication is performed across runs and across models using 100% sequence identity. For the real-world dataset, for each target, the data is de-duplicated, randomly split into three equal parts, and within each part, sequences are ranked by experimentally measured MIC values from low to high, selecting the top-k (k=30) as that part's representative set. This produces 3×30 real-world sequences per target, matching the batch structure of the generative baselines and enabling a fair comparison.

# F INDEPENDENT VALIDATION OF MAC-AMP–GENERATED *E. coli* PEPTIDES

## F.1 INDEPENDENT TESTING OF MAC-AMP GENERATED ANTI-*E. coli* AMP ACTIVITY

To further validate the 90 MAC-AMP–generated anti-*E. coli* AMPs, an external MIC predictor, APEX 1.1, was used. APEX is an ensemble deep-learning model that combines a peptide-sequence encoder with neural predictors of antimicrobial activity (Wan et al., 2024). It was first applied to assess whether each peptide is predicted to be active against at least one of three *E. coli* strains: *E. coli ATCC 11775*, *E. coli AIC221*, and *E. coli AIC222*. Of the 90 AMPs, 85 were predicted to be active against all three *E. coli* strains and 5 were predicted to be active against two of the three *E. coli* strains. Being active was defined as having an MIC $\leq 128$ $\mu$mol l$^{-1}$, as defined in Wan et al. (2024).

## F.2 INDEPENDENT TESTING OF MAC-AMP GENERATED ANTI-*E. coli* AMP BROAD-SPECTRUM ACTIVITY

APEX 1.1 was then used to evaluate broad-spectrum activity across additional clinically relevant pathogens, including *Acinetobacter baumannii* (ATCC 19606), *Klebsiella pneumoniae* (ATCC 13883), *Pseudomonas aeruginosa* (PA01 and PA14), *Staphylococcus aureus* (ATCC 12600 and BAA-1556), and vancomycin-resistant *Enterococcus faecium* (ATCC 700221). These strains cover major ESKAPE pathogens. As shown in Figure A2, many of the generated AMPs exhibit low MIC values across multiple species, indicating broad-spectrum antimicrobial potential.

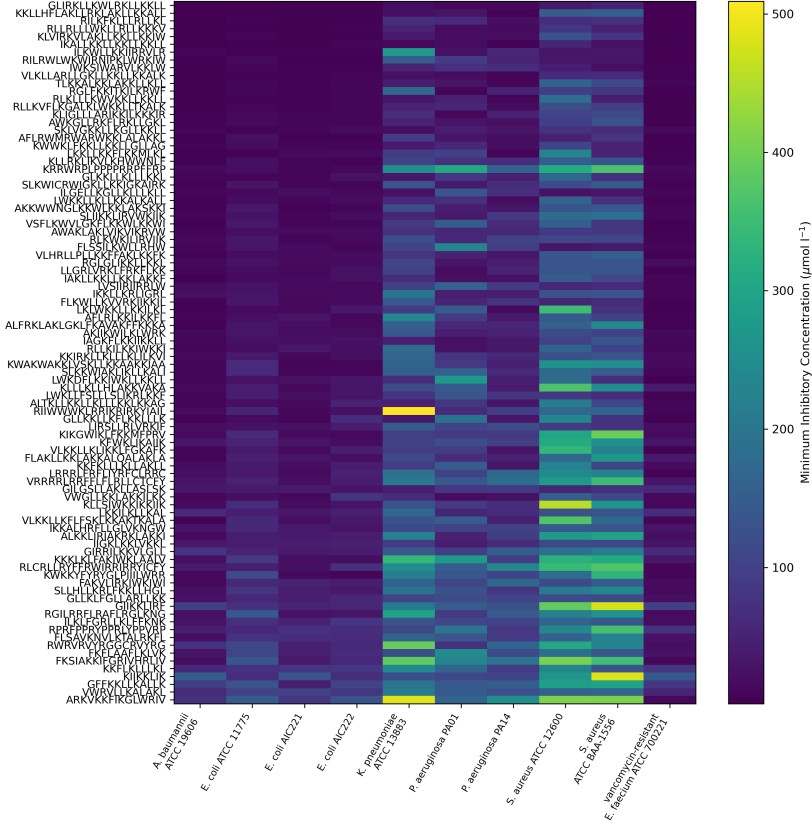

Figure A2: Predicted Minimum Inhibitory Concentrations (MICs) of the top 90 MAC-AMP-generated anti-*Escherichia coli (E. coli)* peptides against *E. coli* strains and ESKAPE pathogens using APEX 1.1.

### F.3 Broad-Spectrum Motif Analysis of MAC-AMP Generated Anti-*E. coli* AMPs

Using a MIC cutoff of $\leq 128\ \mu\mathrm{mol\,L^{-1}}$ to define activity (Wan et al., 2024), the results from broad-spectrum testing with APEX were binarized into active versus inactive for each strain. Among the top 90 MAC-AMP–generated anti-*E. coli* AMPs, 36 (40%) were classified as broad-spectrum, defined as exhibiting activity against at least one strain of each of the five additional ESKAPE pathogens (*A. baumannii, K. pneumoniae, P. aeruginosa, S. aureus, E. faecium*).

For these broad-spectrum AMPs, motif analysis was subsequently performed to investigate potential sequence features underlying their broad-spectrum activity. Motif analysis was performed using MEME 5.5.8 (Bailey et al., 2009) in protein mode under the zero-or-one-occurrence (ZOOPS) model, searching for up to 10 motifs with widths ranging from 3–10 amino acids, using dataset-derived background amino-acid frequencies, Dirichlet mixture priors, and default EM optimization parameters to identify statistically enriched motifs across the AMP set. FIMO 5.5.8 (Grant et al., 2011) was then used to scan all AMP sequences for motif occurrences using the MEME-generated motif file, with a p-value threshold of $1 \times 10^{-3}$ and default settings.

The top ten most frequent motifs identified across the broad-spectrum AMPs are shown in Figure A3. From this, two particularly notable conserved motifs were identified: KFLKGA and WLLGKW. The KFLKGA motif, although not experimentally validated in the literature exactly, exhibits an alternating pattern of cationic (K) and hydrophobic (F, L, A) residues characteristic of amphipathic AMPs and fits the cationic–hydrophobic pattern typical of cationic AMPs. Similarly, the WLLGKW motif follows the same fundamental design principles, featuring a central lysine (K) residue flanked by hydrophobic tryptophan (W) and leucine (L) residues that create an amphipathic structure. The two tryptophan residues are particularly significant, as their large aromatic structures preferentially position at the membrane-water interface, potentially enhancing membrane-disrupting activity. Both motifs are expected to assist the AMPs in targeting bacteria through the characteristic two-step mechanism: initial electrostatic attraction between the cationic lysine residues of the AMP and the negatively charged bacterial membrane surface, followed by hydrophobic insertion of tryptophan and leucine residues into the lipid bilayer. The resulting membrane perturbation and disruption lead to bacterial cell death (Hollmann et al., 2016; Yeaman & Yount, 2003). Overall, this goes to show that there is the potential that the alternating cationic and hydrophobic pattern contributes to the broad-spectrum activity of the generated broad-spectrum AMPs.

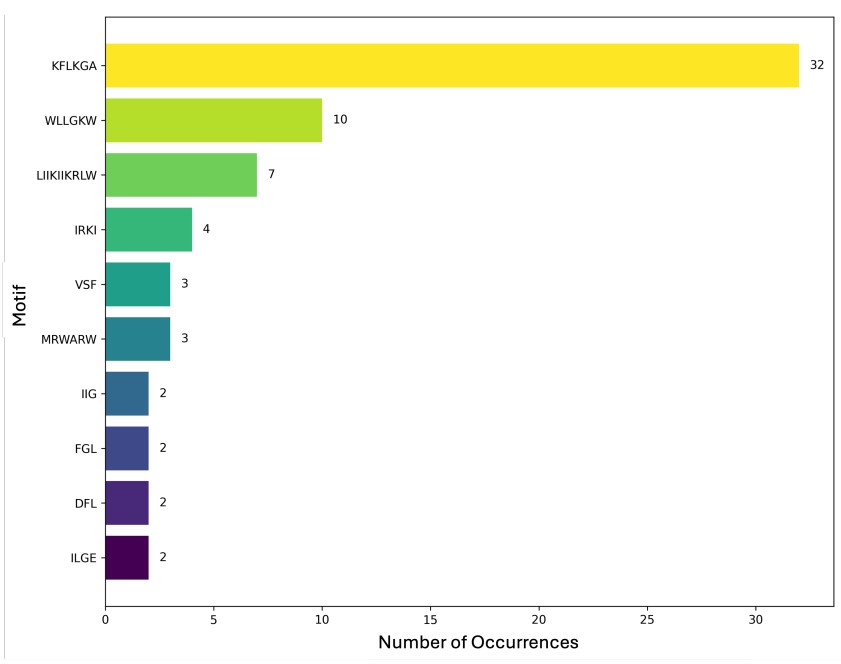

Figure A3: Frequency of the top ten motifs identified in broad-spectrum MAC-AMP–generated anti-*Escherichia coli* antimicrobial peptides.

## G   Novelty of *E. coli* AMPs Generated by MAC-AMP

To ensure novelty, sequence similarity was quantified between the MAC-AMP–generated *E. coli* AMPs and the *E. coli* AMPs in our training dataset (3,818 experimentally validated AMPs from DBAASP v3 and dbAMP 3.0). The Needleman-Wunsch global alignment algorithm implemented in Biopython 1.8.6 was used with default parameters. For each pairwise comparison, similarity was calculated by normalizing the alignment score to the length of the longer peptide in the pair.

Because each generated peptide required comparison against thousands of database sequences, we report, for each generated AMP, the highest and average similarity score observed across all comparisons. As shown in Figure A4, the generated AMPs only show a maximum similarity score of 84.6% to existing AMPs, indicating that even the most similar generated AMPs retain approximately 15% sequence divergence. On average, their similarity to the training dataset sits around 27%, indicating substantial sequence-level differences from known AMPs. Thus, the generated AMPs exhibit consistently low similarity to known AMPs, indicating high sequence-level novelty.

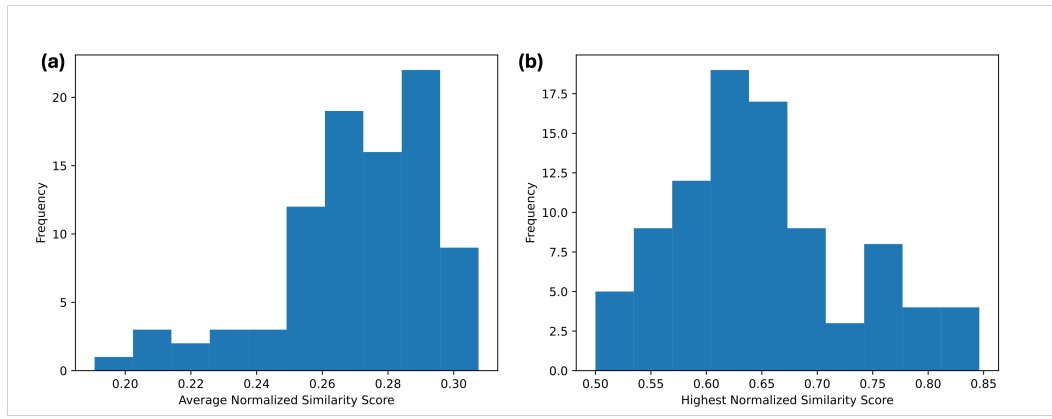

Figure A4: Distribution of (a) average and (b) highest normalized Needleman–Wunsch global-alignment similarity scores for the top 90 MAC-AMP–generated anti-*Escherichia coli (E. coli)* antimicrobial peptides (AMPs), evaluated against an external dataset of real-world *E. coli* AMPs.

The internal similarity among the 90 MAC-AMP–generated *E. coli* AMPs are also assessed using the same alignment procedure described above. Pairwise global alignments were computed for all combinations of generated AMPs, and similarity scores were normalized by division of the maximum sequence length. As shown in Figure A5, internal similarity across the generated set remains low, indicating that MAC-AMP produces a diverse set of AMP sequences rather than minor variations of a few amino acids.

## H   Presence of Antibacterial Activity Related Motifs in *E. coli* AMPs generated by MAC-AMP

To further validate the MAC-AMP results *in silico*, the top 90 anti-*E. coli* AMPs generated by MAC-AMP were analyzed for the presence of motifs experimentally associated with antibacterial activity, particularly against *E. coli*.

A search was conducted for the Cholesterol-Recognizing Amino-Acid Consensus (CRAC) motif and its reverse (CARC), which interact with cholesterol in cell membranes, influence cholesterol-dependent cellular processes, and modulate membrane permeability and stability. CRAC motifs may contribute to antimicrobial activity by sequestering cholesterol or sterol-like lipids, disrupting membrane organization, and forming pores in bacterial membranes. CRAC-containing peptides have demonstrated activity against *E. coli* in previous studies, with CRAC motifs appearing necessary for this effect (Koksharova et al., 2022). Within the generated AMPs, 8 of 90 contained CRAC or CARC motifs, providing additional evidence for activity against *E. coli*.

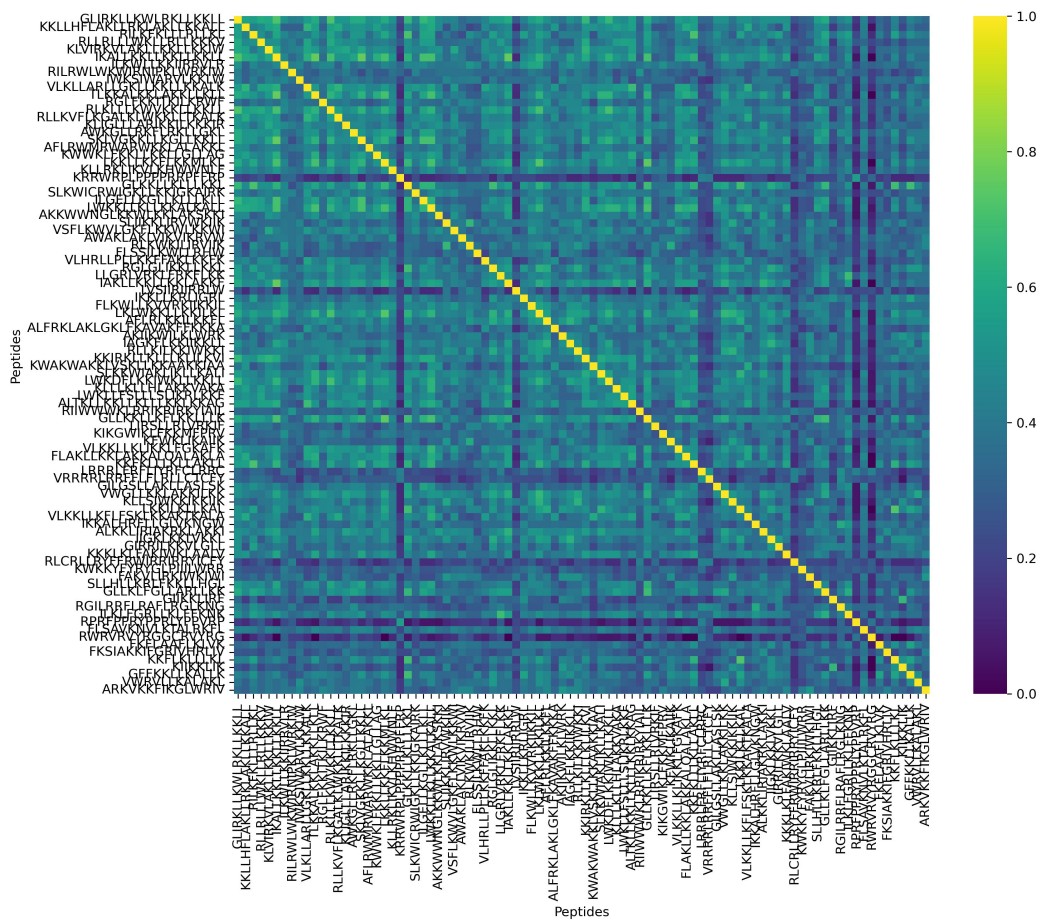

Figure A5: Normalized Needleman-Wunsch global-alignment–based similarity scores between top 90 anti-*Escherichia coli* generated antimicrobial peptides by MAC-AMP.

Another example searched for was proline-rich AMPs (PR-AMPs), which are AMPs with an unusually high content of proline residues and a net cationic charge derived mainly from arginine residues. These peptides have been shown in the literature to demonstrate antibacterial activity. Two AMPs from the generated sequence set are proline-rich: KRRWRPLPPPPRRPFFRP and RPRFPPRYP-PRLYPPVRP. Based on the literature, these peptides likely exhibit anti-*E. coli* activity due to their characteristic proline-rich and arginine-rich composition, a hallmark of PR-AMPs that selectively target Gram-negative bacteria. The first AMP (KRRWRPLPPPPRRPFFRP) contains an N-terminal arginine-rich region, which has been shown to influence antimicrobial activity, along with multiple proline residues arranged in patterns similar to functional PR-AMP fragments. The second AMP (RPRFPPRYPPPRLYPPVRP) displays repeating Pro-Arg motifs (particularly PRP sequences) that are characteristic of active PR-AMPs (Scocchi et al., 2011).

Overall, this provides some evidence into the mechanisms and validity of the AMPs designed by MAC-AMP to target *E. coli in silico*.

# I GENERAL STRUCTURAL STABILITY OF *E. coli* AMPS GENERATED BY MAC-AMP

OmegaFold pLDDT is used as a structural reliability proxy to guide MAC-AMP during peptide generation. To confirm that MAC-AMP produces generally stable peptides and that OmegaFold pLDDT is a reliable proxy, molecular dynamics (MD) simulations were performed on a subset

of randomly chosen *E. coli* AMPs from the top 90 candidates generated by MAC-AMP to assess general structural stability.

MD simulations were performed using OpenMM 8.4.0 (Eastman et al., 2024) to assess the structural stability of the generated AMP structures. Input Protein Data Bank (PDB) files were first processed using PDBFixer 1.12 to add missing residues, missing heavy atoms, and hydrogens at pH 7.0. Each prepared structure was then solvated in a TIP3P water box with 0.8 nm padding and 0.15 M ionic strength using the AMBER14 force field. The systems were energy-minimized, followed by 3 ns of NPT equilibration at 300 K and 1 bar using a Langevin integrator with a 2 fs timestep and a Monte Carlo barostat. Production MD simulations were conducted for 100 ns with coordinates saved every 1 ps. Backbone Root Mean Square Deviation (RMSD) values were calculated using MDTraj 1.11 (McGibbon et al., 2015), which measures the average distance between the backbone atoms of a peptide structure over time compared to the initial structure. General conditions were used to simply assess the reliability of OmegaFold to guide the design of AMPs with structural reliability.

Table A1 shows that the mean RMSD lies around 2-4 Åfor the AMPs generated by MAC-AMP. This shows that the AMPs designed by MAC-AMPs demonstrate general structural stability and also that using OmegaFold pLDDT as a proxy to guide the generation of AMPs is acceptable.

Table A1: Backbone root mean square deviation (RMSD) of molecular dynamics simulations of MAC-AMP-generated anti-*Escherichia coli* antimicrobial peptides (AMPs) (mean ± standard deviation)

| AMP | RMSD (Å) | AMP | RMSD (Å) | AMP | RMSD (Å) | AMP | RMSD (Å) |
|---|---|---|---|---|---|---|---|
| 1 | 2.22 ± 0.79 | 17 | 3.18 ± 1.49 | 33 | 2.50 ± 0.79 | 49 | 2.57 ± 0.67 |
| 2 | 2.93 ± 0.97 | 18 | 2.25 ± 0.94 | 34 | 3.66 ± 0.64 | 50 | 2.09 ± 0.72 |
| 3 | 2.16 ± 1.12 | 19 | 1.61 ± 0.38 | 35 | 2.68 ± 1.00 | 51 | 2.61 ± 0.70 |
| 4 | 3.40 ± 1.85 | 20 | 3.27 ± 0.86 | 36 | 1.88 ± 0.48 | 52 | 4.68 ± 1.20 |
| 5 | 2.17 ± 0.68 | 21 | 2.17 ± 0.64 | 37 | 3.65 ± 1.25 | 53 | 3.63 ± 1.84 |
| 6 | 2.66 ± 0.53 | 22 | 1.42 ± 0.54 | 38 | 3.70 ± 1.23 | 54 | 2.51 ± 0.49 |
| 7 | 4.04 ± 1.61 | 23 | 3.32 ± 0.62 | 39 | 3.07 ± 0.98 | 55 | 3.85 ± 0.83 |
| 8 | 2.91 ± 0.55 | 24 | 1.73 ± 0.57 | 40 | 3.53 ± 1.00 | 56 | 2.54 ± 0.81 |
| 9 | 2.66 ± 0.51 | 25 | 1.85 ± 0.65 | 41 | 2.23 ± 1.14 | 57 | 4.59 ± 0.91 |
| 10 | 2.87 ± 0.72 | 26 | 2.50 ± 0.63 | 42 | 3.26 ± 1.45 | 58 | 1.68 ± 0.70 |
| 11 | 3.63 ± 0.88 | 27 | 3.15 ± 0.71 | 43 | 2.67 ± 1.38 | 59 | 2.81 ± 0.91 |
| 12 | 2.85 ± 0.65 | 28 | 2.70 ± 0.86 | 44 | 4.38 ± 0.81 | 60 | 1.91 ± 0.49 |
| 13 | 3.84 ± 1.29 | 29 | 1.80 ± 0.64 | 45 | 2.23 ± 0.65 | 61 | 2.65 ± 0.69 |
| 14 | 2.87 ± 0.45 | 30 | 1.79 ± 1.04 | 46 | 4.00 ± 1.03 | 62 | 1.63 ± 0.68 |
| 15 | 4.96 ± 1.19 | 31 | 1.51 ± 0.79 | 47 | 3.79 ± 1.46 | 63 | 3.47 ± 0.80 |
| 16 | 2.91 ± 0.99 | 32 | 3.21 ± 1.31 | 48 | 2.32 ± 1.15 | 64 | 3.36 ± 0.84 |

## J    SEQUENCE-LEVEL AND BIOPHYSICAL PROPERTIES OF *E. coli* AMPS GENERATED BY MAC-AMP

Here, sequence-level (Table A2) and biophysical (Table A3) features are summarized for six generated AMPs by MAC-AMP for *E. coli* inhibition. In Table A2, the Length column indicates the number of residues in the peptide sequence. The K/R Fraction column reports the fraction of lysine (K) and arginine (R) residues. The K/R Fraction Positions column shows the positions of K and R residues within each sequence. The Identity Fraction gives the proportion of residues identical to the reference template sequence used during generation. In Table A3, GRAVY represents the Grand Average of Hydropathy, quantifying the overall hydrophobicity of the peptide sequence. Hydrophobic Moment ($\alpha$-helix) measures the $\alpha$-helical amphipathicity, indicating the degree of segregation between hydrophobic and hydrophilic faces when the peptide adopts an $\alpha$-helical conformation. Net Charge at pH 7 indicates the total charge of the peptide under physiological conditions. pI gives the isoelectric point, which is the pH at which the peptide carries no net charge.

The generated peptides exhibit key characteristics associated with antimicrobial activity: moderate to high cationic character (K/R fractions of 0.32-0.41, net charges of +6 to +9), balanced to moderately high hydrophobicity (GRAVY values of 0.16-0.57), and moderate to strong amphipathicity

(hydrophobic moments of 0.38-0.73). Notably, all sequences maintain high pI values (11.25-12.97), ensuring they remain positively charged under physiological conditions, which facilitates electrostatic interaction with negatively charged bacterial membranes. The identity fractions (0.50-0.60) indicate that MAC-AMP modified 40-50% of residues relative to template sequences to optimize antimicrobial properties.

Table A2: Summary of sequence-level features for six generated antimicrobial peptides by MAC-AMP for *Escherichia coli*

| Sequence | Length | K/R Fraction | K/R Fraction Positions | Identity Fraction |
|---|---|---|---|---|
| FRVFGFIAKKVKKLVKKI | 18 | 0.389 | 1,8,9,11,12,15,16 | 0.556 |
| VRGGAIKKIAKILAKLLAR | 19 | 0.316 | 1,6,7,10,14,18 | 0.579 |
| VGLVKKWFKSVIKKVAKS | 18 | 0.333 | 4,5,8,12,13,16 | 0.500 |
| RIFKFLKRAFGIIGLFKRRIKS | 22 | 0.364 | 0,3,6,7,16,17,18,20 | 0.500 |
| KIWKLLKKVLAKVAK | 15 | 0.400 | 0,3,6,7,11,14 | 0.600 |
| IIGKLVLKKVGKIIKKILKKKA | 22 | 0.409 | 3,7,8,11,14,15,18,19,20 | 0.500 |

Table A3: Summary of biophysical properties for six generated antimicrobial peptides by MAC-AMP for *Escherichia coli*

| Sequence | GRAVY | Hydrophobic Moment$_{\alpha\text{-helix}}$ | Net Charge$_{pH7}$ | pI |
|---|---|---|---|---|
| FRVFGFIAKKVKKLVKKI | 0.406 | 0.734 | 7 | 11.95 |
| VRGGAIKKIAKILAKLLAR | 0.574 | 0.375 | 6 | 12.52 |
| VGLVKKWFKSVIKKVAKS | 0.189 | 0.546 | 6 | 11.25 |
| RIFKFLKRAFGIIGLFKRRIKS | 0.155 | 0.578 | 8 | 12.97 |
| KIWKLLKKVLAKVAK | 0.240 | 0.721 | 6 | 11.25 |
| IIGKLVLKKVGKIIKKILKKKA | 0.373 | 0.427 | 9 | 11.45 |

# K    COMPARATIVE ANALYSIS OF SEQUENCE AND BIOPHYSICAL PROPERTIES OF *E. coli* AMPS GENERATED BY MAC-AMP VS. BASELINE MODELS

To evaluate sequence and biophysical properties, 2,000 random AMPs were generated by MAC-AMP for *E. coli* and compared with equal-sized sets from baseline models and the training dataset of experimentally validated *E. coli* AMPs (Figure A6). Overall, MAC-AMP generates AMPs with distributions similar to experimentally validated anti-*E. coli* AMPs while showing enhanced optimization for key properties associated with antimicrobial activity. In Figure A6a, MAC-AMP's amino acid composition shows elevated K/R and L/I, consistent with cationic, amphipathic $\alpha$-helices, while remaining broadly similar to real AMPs. Figure A6b illustrates that MAC-AMP peptides peak at a global charge of +5–7 at pH 7.4, reflecting a design preference for increased membrane interaction. GRAVY distributions (Figure A6c) indicate moderate hydrophobicity (median $\sim$0 to +0.5), slightly more hydrophobic than natural AMPs, which may enhance membrane insertion potential while maintaining solubility. Predicted helix fractions (Figure A6d) are slightly elevated ($\sim$0.45–0.5), supporting formation of stable amphipathic helices. Sequence lengths (Figure A6e) are predominantly 17–20 aa, well within the typical 10–40 aa AMP range. Also, Eisenberg hydrophobic moments (Figure A6f) are highest among all models ($\sim$0.55 median), producing well-defined amphipathic faces without extreme hydrophobic clustering, which could lead to aggregation or reduced solubility. Finally, the instability index (Figure A6g) and global hydrophilicity (Figure A6h) distributions are comparable to experimentally validated AMPs, indicating that MAC-AMP maintains stability and solubility profiles consistent with functional peptides. Collectively, this indicates that MAC-AMP maintains the statistical fidelity of natural AMPs while optimizing charge, amphipathicity, and helicity to favour membrane-active designs.

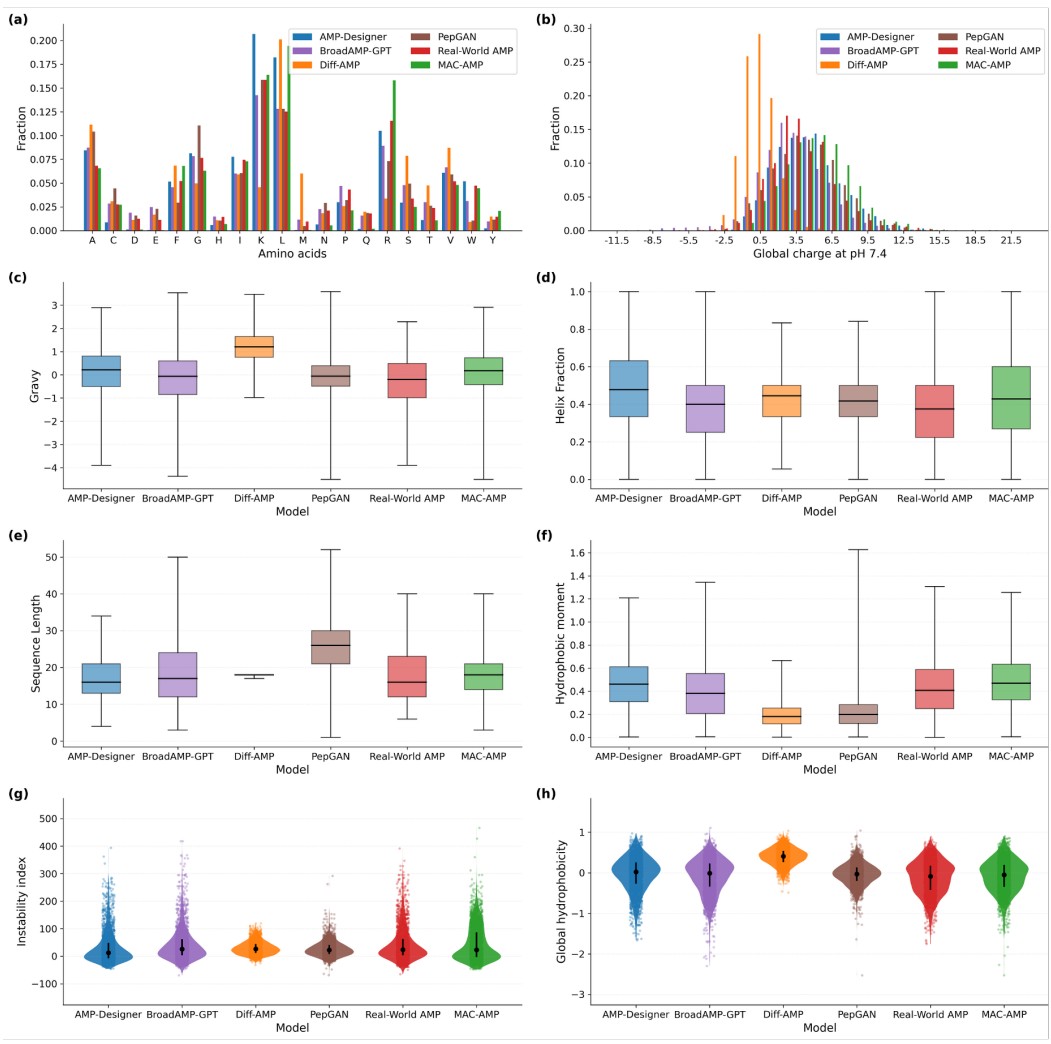

Figure A6: Comparative distributions of sequence and biophysical properties across 2000 random antimicrobial peptides (AMPs) generated by MAC-AMP, five baseline models, and a real-world AMP dataset targeting *Escherichia coli*. (a) Amino-acid composition of AMPs, normalized to per-model fractions. (b) Global charge at pH 7.4 for all AMPs. (c) Grand Average of Hydropathy (GRAVY) of AMPs, representing overall hydrophobicity. (d) Predicted helix fraction of AMPs, indicating their propensity to adopt $\alpha$-helical structure. (e) AMPs length in amino acids. (f) Eisenberg hydrophobic moment of AMPs, reflecting amphipathicity. (g) Instability index of AMPs, estimating their predicted stability. (h) Global hydrophobicity of AMPs.

## L    ABLATION STUDIES

### L.1    ABLATION STUDIES ON PROPERTY PREDICTION MODULE

Ablation studies were conducted on the Property Prediction module by comparing MAC-AMP to versions where each of the following additional objectives was removed in the Property Prediction module: Toxicity ($V_a$), Structural Reliability ($V_b$), and AMP Likelihood ($S_b$).

The performance metrics of the generated AMPs by each variant compared to MAC-AMP are summarized in Table A4. The computational costs were also measured via GPU hours, total number of API calls, peak memory usage in MB, and API costs, shown in Figure A7. As expected, although computational costs drop when multiple objectives are removed, it comes at the cost of performance. When each or a combination of objectives is removed, the model can optimize for the remaining ob-

jectives (instead of having to balance all three), so it produces slightly higher performance metrics for those optimized remaining objectives at the cost of the other objectives. The slight increase noted in the optimized objectives when others are dropped is not worth the trade-off of the multi-objective optimization. Overall, MAC-AMP can effectively optimize AMP generation for multiple objectives at once when all components are included.

Table A4: Ablation studies of the Property Prediction module: property scores of antimicrobial peptides generated by MAC-AMP and its variants. Here, $V_a$ denotes the toxicity predictor (ToxinPred 3.0), $V_b$ the structural-reliability property predictor (OmegaFold v1), and $S_b$ the AMP-likelihood predictor (Macrel 1.5).

| Model | Antibacterial Activity (↑) | AMP Likelihood (↑) | Toxicity (↓) | Structural Reliability (↑) |
|---|---|---|---|---|
| MAC-AMP | $0.943 \pm 0.008$ | $0.797 \pm 0.012$ | $0.154 \pm 0.008$ | $0.873 \pm 0.009$ |
| Drop_$V_b$ | $0.904 \pm 0.028$ | $0.799 \pm 0.009$ | $0.157 \pm 0.021$ | $0.818 \pm 0.012$ |
| Drop_$V_a$ | $0.946 \pm 0.045$ | $0.782 \pm 0.016$ | $0.231 \pm 0.028$ | $0.799 \pm 0.017$ |
| Drop_$S_b$ | $0.923 \pm 0.037$ | $0.742 \pm 0.006$ | $0.164 \pm 0.010$ | $0.830 \pm 0.014$ |
| Drop_$V_a$_$V_b$ | $0.904 \pm 0.043$ | $0.764 \pm 0.023$ | $0.183 \pm 0.016$ | $0.758 \pm 0.015$ |
| Drop_$V_b$_$S_b$ | $0.889 \pm 0.040$ | $0.745 \pm 0.035$ | $0.134 \pm 0.087$ | $0.763 \pm 0.017$ |
| Drop_$V_a$_$S_b$ | $0.933 \pm 0.021$ | $0.735 \pm 0.010$ | $0.255 \pm 0.024$ | $0.836 \pm 0.019$ |
| Drop_$V_a$_$V_b$_$S_b$ | $0.866 \pm 0.044$ | $0.729 \pm 0.024$ | $0.212 \pm 0.045$ | $0.742 \pm 0.031$ |

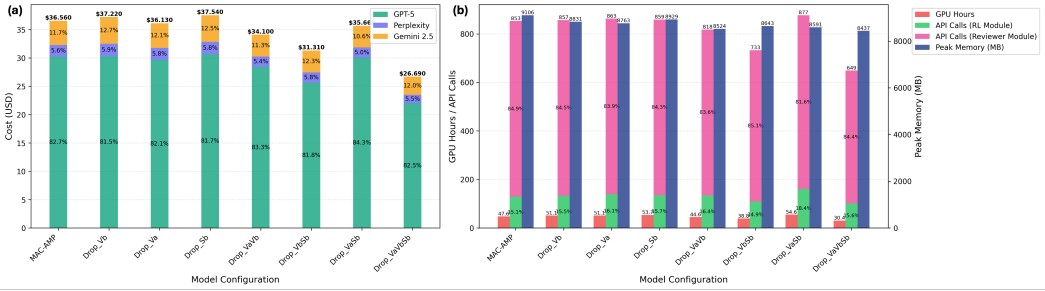

Figure A7: Ablation studies of the Property Prediction module. (a) API costs for each ablation model, broken down by the percentage contribution of each Reviewer Agent: GPT-5, Perplexity, and Gemini 2.5. (b) GPU hours, total number of API calls (further separated into percentages from the Reinforcement Learning Refinement module versus the Artificial Intelligence-simulated Peer Reviewer module), and peak memory usage (MB) for each ablation model. Here, $V_a$ denotes the toxicity predictor (ToxinPred 3.0), $V_b$ the structural-reliability property predictor (OmegaFold v1), and $S_b$ the AMP-likelihood predictor (Macrel 1.5).

## L.2 ABLATION STUDIES ON AI-SIMULATED PEER REVIEW MODULE

Ablation studies were conducted on the contribution of the AI-simulated Peer Review module by comparing MAC-AMP to versions where each of the Reviewer agents was removed in the AI-simulated Peer Review module. As shown in Table A5, the AI-Simulated Peer Review module is key to achieving a balanced multi-objective outcome. When all Reviewer agents are removed, and only RL with handcrafted rewards is retained, all four metrics deteriorate. The three Reviewer agents have distinct roles and counterbalance one another. GPT-5 (RA1) functions as the gatekeeper of safety and discriminability. When GPT-5 is included, toxicity scores decrease and AMP scores remain high, although antibacterial activity and structural reliability scores are slightly reduced. Conversely, removing GPT-5 produces candidates that appear more potent but exhibit increased toxicity and reduced stability. Perplexity (RA2) acts as the primary driver of potency. Removing Perplexity reduces all four metrics, whereas using Perplexity alone increases antibacterial activity scores but leads to lower AMP scores, higher toxicity, and weaker structural stability, indicating a unilateral bias when applied in isolation. Gemini 2.5 (RA3) primarily mediates the synergy between structural reliability and potency. Removing Gemini 2.5 results in a pronounced decline in structural

reliability, while using Gemini 2.5 alone fails to balance safety and overall performance. Together, the three Reviewer agents form a closed loop: Perplexity pushes sequences into the effective region, GPT-5 pulls risk back into a controllable range, and Gemini 2.5 improves foldability, ultimately delivering the most balanced results across all four metrics.

Table A5: Ablation studies of the AI-simulated Peer Review module: property scores of antimicrobial peptides generated by MAC-AMP and its variants. Here, RA1 denotes Reviewer agent 1 (GPT-5), RA2 the Reviewer agent 2 (Perplexity), and RA3 the Reviewer agent 3 (Gemini 2.5)

| Model | Antibacterial Activity (↑) | AMP Likelihood (↑) | Toxicity (↓) | Structural Reliability (↑) |
|---|---|---|---|---|
| MAC-AMP | $0.943 \pm 0.008$ | $0.797 \pm 0.012$ | $0.154 \pm 0.008$ | $0.873 \pm 0.009$ |
| w/o RA3 | $0.857 \pm 0.016$ | $0.840 \pm 0.008$ | $0.133 \pm 0.023$ | $0.731 \pm 0.017$ |
| w/o RA2 | $0.884 \pm 0.015$ | $0.780 \pm 0.008$ | $0.164 \pm 0.023$ | $0.734 \pm 0.021$ |
| w/o RA1 | $0.953 \pm 0.010$ | $0.742 \pm 0.016$ | $0.181 \pm 0.019$ | $0.783 \pm 0.017$ |
| w/o RA2 + RA3 | $0.870 \pm 0.013$ | $0.806 \pm 0.006$ | $0.130 \pm 0.015$ | $0.740 \pm 0.011$ |
| w/o RA1 + RA3 | $0.979 \pm 0.010$ | $0.700 \pm 0.019$ | $0.185 \pm 0.019$ | $0.758 \pm 0.015$ |
| w/o RA1 + RA2 | $0.938 \pm 0.011$ | $0.795 \pm 0.008$ | $0.195 \pm 0.013$ | $0.862 \pm 0.013$ |
| w/o All Review | $0.825 \pm 0.016$ | $0.627 \pm 0.014$ | $0.448 \pm 0.051$ | $0.831 \pm 0.012$ |

## L.3 Ablation Studies on Reinforcement Learning Refinement Module

Ablation studies were conducted on the contribution of the RL Refinement module by comparing MAC-AMP to versions where the RL Reward Decision agent and/or Adaptive Optimization were removed or replaced with human-designed RL, where the reward and PPO code were designed by a human expert. In this set of RL module ablations, MAC-AMP, with both the RL Reward Decision agent and Adaptive Optimization, achieves the most balanced performance across all metrics (Table A6). In more detail, without the RL Reward Decision agent, all property metrics drop except for structural reliability, which slightly increases, leading to a clear safety degradation. Without adaptive optimization, antibacterial activity superficially increases while all other properties drop, reflecting overexploitation under a fixed-reward and PPO strategy. As such, it is clear that the stagewise updates act as regularization and calibration, preventing the search from being pulled too hard by one specific objective. Without either, performance degrades across the board except for toxicity, likely because the static reward places a heavy penalty on toxicity. In short, the RL Reward Decision agent determines whether the right trade-off was chosen, and Adaptive Optimization determines whether it is stable. Both are indispensable for reaching a balanced multi-objective optimum.

Table A6: Ablation studies of the Reinforcement Learning Refinement module: property scores of antimicrobial peptides generated by MAC-AMP and its variants

| Model | Antibacterial Activity (↑) | AMP Likelihood (↑) | Toxicity (↓) | Structural Reliability (↑) |
|---|---|---|---|---|
| MAC-AMP | $0.943 \pm 0.008$ | $0.797 \pm 0.012$ | $0.154 \pm 0.008$ | $0.873 \pm 0.009$ |
| Without RL Decision Agent | $0.935 \pm 0.011$ | $0.791 \pm 0.013$ | $0.260 \pm 0.022$ | $0.879 \pm 0.010$ |
| Without Adaptive Optimization | $0.955 \pm 0.003$ | $0.752 \pm 0.005$ | $0.279 \pm 0.006$ | $0.869 \pm 0.013$ |
| Without RL Decision Agent & Adaptive Optimization | $0.818 \pm 0.021$ | $0.737 \pm 0.009$ | $0.200 \pm 0.022$ | $0.768 \pm 0.020$ |
| Replaced by Human-Designed RL | $0.900 \pm 0.014$ | $0.776 \pm 0.009$ | $0.175 \pm 0.018$ | $0.759 \pm 0.015$ |

# M    SUBSTITUTION ANALYSES

To validate our model's performance, substitution analyses were conducted where different components of our model framework were swapped, and computational costs and/or performance were evaluated in comparison to the original framework. For each analysis, downstream performance was measured via the generated AMPs' antibacterial activity, AMP likelihood, toxicity and structural reliability. When computational costs were calculated, GPU hours, total number of API calls (broken down by the percentage coming from the RL Refinement module versus the AI-simulated Peer Review module), peak memory usage, and API costs (broken down by the percentage contributed by each Reviewer agent) were noted.

## M.1    SUBSTITUTION ANALYSIS ON NUMBER OF EXTERNAL RL TRAINING EPOCHS

In MAC-AMP, the RL Refinement module is trained for 15 epochs by default. To evaluate the effect of training duration, the number of epochs was reduced to 10 and increased to 20, and computational costs are reported in Figure A8. Training for 15 epochs incurs only an additional $0.46 USD compared to 10 epochs, whereas increasing to 20 epochs raises API costs by $5.62 USD. GPU hours increase with the number of epochs, while peak memory usage remains largely unchanged. To note, MAC-AMP allows users to choose their desired number of external RL training epochs depending on their preferred balance between accuracy and computational cost.

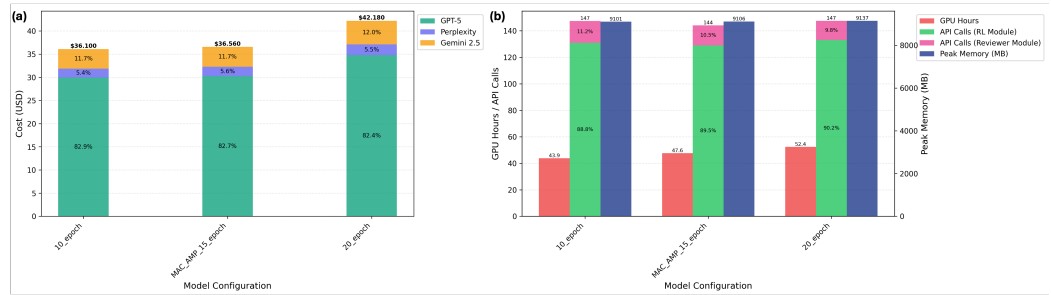

Figure A8: Substitution analyses on the number of Reinforcement Learning module training epochs. (a) API costs for each substitution variant, broken down by the percentage contribution of each Reviewer Agent: GPT-5, Perplexity, and Gemini 2.5. (b) GPU hours, total number of API calls (further separated into percentages from the Reinforcement Learning module versus the AI-simulated Peer Review module), and peak memory usage (MB) for each substitution variant.

## M.2    SUBSTITUTION ANALYSIS ON MAXIMUM GENERATED PEPTIDE LENGTH

In MAC-AMP, the default maximum peptide length is 32, reflecting a focus on designing short AMPs. The effects of lowering the maximum length to 20 and 26 or increasing it to 39 were explored, and as shown in Figure A9, the computational costs due to differences in peptide length are negligible, so sequence length does not have a significant impact on computational costs.

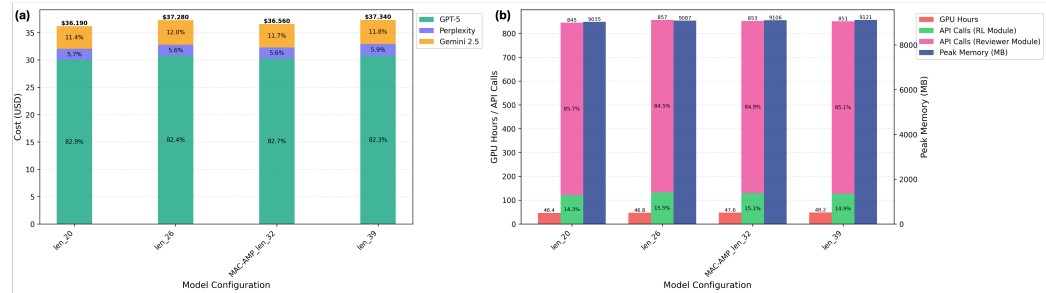

Figure A9: Substitution analyses on the maximum length of generated peptides. (a) API costs for each substitution variant, broken down by the percentage contribution of each Reviewer Agent: GPT-5, Perplexity, and Gemini 2.5. (b) GPU hours, total number of API calls (further separated into percentages from the Reinforcement Learning module versus the Artificial Intelligence-simulated Peer Review module), and peak memory usage (MB) for each substitution variant.

## M.3 SUBSTITUTION ANALYSIS ON AI-SIMULATED PEER REVIEW MODULE AGENTS

MAC-AMP currently uses GPT-5, Perplexity, and Gemini 2.5 as Reviewer agents and the Area Chair agent. To evaluate cost differences, the agents were replaced with locally deployed small models in four configurations: RA_AC_Llama, in which all Reviewer agents and the Area Chair agent in the Peer Review module were replaced with Llama 3.1 8B (Grattafiori & Dubey, 2024); RA_AC_Qwen, in which all Reviewer agents and the Area Chair agent were replaced with Qwen 2.5 7B-instruct (Qwen et al., 2024); RA_API_AC_Qwen, in which only the Area Chair agent was replaced with Qwen; and RA_Qwen_AC_API, in which only the Reviewer agents were replaced with Qwen. As shown in Table A7, MAC-AMP achieves the most favourable balance across all evaluated metrics, particularly excelling in toxicity (0.154) and AMP likelihood (0.797), while maintaining competitive antibacterial activity and structural reliability. Variants that obtain higher antibacterial activity or structural reliability do so at the cost of substantially increased toxicity (2–3× worse) and reduced AMP likelihood. Because all objectives are jointly prioritized, the MAC-AMP agents represent the most effective choice overall. Although MAC-AMP incurs the highest API and computational costs (with the exception of peak memory), as shown in Figure A10, performance is prioritized over computational efficiency in this work, and the resulting costs remain acceptable for the intended use. Therefore, MAC-AMP remains the preferred configuration.

Table A7: Substitution analyses on the Artificial Intelligence-simulated Peer Review Module Agents: property scores of antimicrobial peptides generated by MAC-AMP and its variants. RA_AC_Llama: all Reviewer agents (RA) and the Area Chair (AC) agent are replaced with Llama 3.1 8B, RA_AC_Qwen: all RA and the AC agent are replaced with Qwen 2.5 7B-instruct, RA_API_AC_Qwen: only the AC agent was replaced with Qwen, RA_Qwen_AC_API: only the RA are replaced with Qwen

| Model | Antibacterial Activity (↑) | AMP Likelihood (↑) | Toxicity (↓) | Structural Reliability (↑) |
|---|---|---|---|---|
| MAC-AMP | $0.943 \pm 0.008$ | $0.797 \pm 0.012$ | $0.154 \pm 0.008$ | $0.873 \pm 0.009$ |
| RA_AC_Llama | $0.944 \pm 0.007$ | $0.713 \pm 0.012$ | $0.201 \pm 0.020$ | $0.896 \pm 0.012$ |
| RA_AC_Qwen | $0.949 \pm 0.009$ | $0.683 \pm 0.010$ | $0.308 \pm 0.013$ | $0.913 \pm 0.016$ |
| RA_API_AC_Qwen | $0.865 \pm 0.014$ | $0.783 \pm 0.011$ | $0.206 \pm 0.013$ | $0.863 \pm 0.007$ |
| RA_Qwen_AC_API | $0.835 \pm 0.023$ | $0.734 \pm 0.017$ | $0.211 \pm 0.030$ | $0.854 \pm 0.015$ |

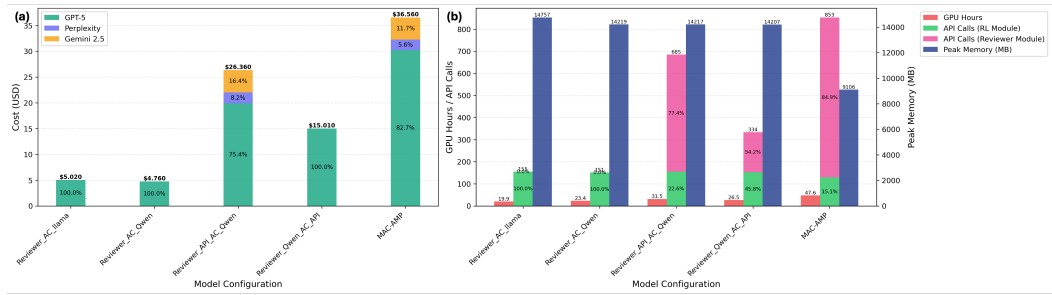

Figure A10: Substitution analyses on the AI-simulated Peer Review Module Agents. (a) API costs for each comparative model, broken down by the percentage contribution of each Reviewer Agent: GPT-5, Perplexity, and Gemini 2.5. (b) GPU hours, total number of API calls (further separated into percentages from the Reinforcement Learning module versus the Artificial Intelligence-simulated Peer Review module), and peak memory usage (MB) for each comparative model.

### M.4 SUBSTITUTION ANALYSIS ON TOXICITY PROPERTY PREDICTION MODEL

For MAC-AMP, ToxinPred 3.0 is used as the toxicity property prediction model. At the time of ToxinPred 3.0's release, it remained one of the top general toxicity prediction models. CAPTP was tested as an alternative to determine whether ToxinPred 3.0 is the superior choice. CAPTP is a deep learning model that combines convolutional and self-attention mechanisms, using convolutional layers to extract local sequence motifs, self-attention layers to capture long-range dependencies across the peptide, and fully connected layers to integrate these learned features for highly accurate toxicity prediction (Jiao et al., 2024).

ToxinPred 3.0 and CAPTP were both evaluated on an independent dataset originally used to test ToxinPred 3.0, ensuring that none of the compounds overlapped with the training sets of either model. As detailed in Table A8, ToxinPred 3.0 outperforms CAPTP.

Table A8: Performance comparison between ToxinPred 3.0 and CAPTP

| Predictor | Accuracy | Weighted Precision | Weighted Recall | Weighted F1-score |
|---|---|---|---|---|
| ToxinPred 3.0 | 0.828 | 0.849 | 0.828 | 0.828 |
| CAPTP | 0.770 | 0.787 | 0.77 | 0.770 |

### M.5 SUBSTITUTION ANALYSIS ON MIC PROPERTY PREDICTION MODEL

For MAC-AMP, we designed an LLM-based regressor (MAC-AMP MIC Predictor), adapted from BERT AmPEP60 (Cai et al., 2025), that fine-tunes ProtBERT via transfer learning (Elnaggar et al., 2022) to predict species-specific MIC values for AMP sequences. The designed MIC prediction model was compared to APEX 1.1, an ensemble of deep-learning networks that uses a peptide-sequence encoder coupled with neural predictors of antimicrobial activity (Wan et al., 2024).

Both models were evaluated on a collection of 157 AMPs targeting *E. coli ATCC 11775*, *E. coli AIC221*, and/or *E. coli AIC222* from DBAASP v3, after removing duplicates and any overlap with our training dataset. As shown in Table A9, our predictor outperforms APEX. Therefore, using our self-trained MIC predictor within the framework is the optimal choice.

Table A9: Performance comparison between MAC-AMP MIC Predictor and APEX 1.1

| Predictor | $R^2$ | Pearson $r$ | Spearman $\rho$ |
|---|---|---|---|
| MAC-AMP MIC Predictor | 0.572 | 0.758 | 0.724 |
| APEX 1.1 | 0.546 | 0.728 | 0.607 |

## M.6    Substitution Analysis on Lexicon Weights

The lexicon weights in the weight-tagging map of the AI-simulated Peer Review module are determined by the agent itself during a preparatory meeting with a human expert, subject to constraints such as restricting the possible distribution of each dimensional score (summary of weights) to the range $[-1, 1]$. This process occurred prior to training and established the injectable knowledge for the model.

Substitution analyses were performed to evaluate the change in performance metrics when the lexicon weights decided by the model are offset by 0.1. As shown in Table A10, decreasing the lexicon weights by 0.1 resulted in a slight negative impact on performance, whereas increasing the weights by 0.1 caused a steep decline. Overall, the weights determined during the human expert–agent preparatory meeting remain reliable and efficient.

Table A10: Substitution analyses on the lexicon weights: property scores of antimicrobial peptides generated by MAC-AMP and its variants

| Model | Antibacterial Activity ($\uparrow$) | AMP Likelihood ($\uparrow$) | Toxicity ($\downarrow$) | Structural Reliability ($\uparrow$) |
|---|---|---|---|---|
| MAC-AMP | $0.943 \pm 0.008$ | $0.797 \pm 0.012$ | $0.154 \pm 0.008$ | $0.873 \pm 0.009$ |
| Lexicon Weight + 0.1 | $0.878 \pm 0.023$ | $0.794 \pm 0.009$ | $0.156 \pm 0.012$ | $0.836 \pm 0.013$ |
| Lexicon Weight - 0.1 | $0.938 \pm 0.019$ | $0.804 \pm 0.006$ | $0.171 \pm 0.012$ | $0.871 \pm 0.013$ |

# N    Cross-Domain Transferability

One of MAC-AMP's biggest novelties is its setup for cross-domain transferability. To test its potential beyond AMP design, the MAC-AMP framework was applied for an English table-to-text generation task, where, given a table and a set of highlighted table cells, the model has to produce a one-sentence description. To test this task, a subset of the ToTTo dataset was used, an open-domain English table-to-text dataset with training examples of Wikipedia tables, highlighted table cells, and one-sentence descriptions (Parikh et al., 2020). For this preliminary study, a randomly sampled subset of 30,000 training examples (approximately 25% of the full dataset of 120,761 sequences) was used.

In transferring MAC-AMP to the ToTTo table-to-text benchmark, the multi-agent architecture, logging scheme, and PPO training code remain unchanged, and only minimal domain-specific substitutions are applied. The Property Prediction module no longer queries molecular predictors; instead, each candidate description was evaluated using the official ToTTo scripts to obtain PARENT and BLEU scores, which are linearly normalized and used as $S_a$ and $S_b$ within the RL pipeline. Additional statistics, such as table coverage, proportion of unsupported tokens, and length ratio, are computed and exposed to the AI-simulated Peer Review module as V-style auxiliary evidence. The module continues to operate over the four dimensions (Eff / Safe / DevStruct / Orig), but their semantics are redefined for text generation. Eff denotes task effectiveness and content adequacy (coverage of highlighted cells and correctness of expressed facts). Safe denotes factual safety (absence of hallucinations or contradictions relative to the table). DevStruct denotes linguistic quality and structural coherence (grammaticality, fluency, and Wikipedia-like discourse organization). Orig denotes stylistic diversity and generalization (faithful yet varied paraphrasing rather than verbatim copying). Based on these redefinitions, a new weighted lexicon is constructed for each dimension, while the original tagging mechanism and scoring procedure remain unmodified.

In the RL Refinement and Peptide Generation modules, the peptide generator is replaced by a T5-small encoder–decoder model. The reward-design and alignment agents receive updated prompts describing PARENT/BLEU and the redefined four dimensions, but still emit TorchScript reward functions, $F(S_a, S_b, S_c) \in [0, 1]$, which are consumed by the same PPO strategy to update the generator.

Due to time limitations, only a minimal evaluation was conducted in which both the T5-based baseline and the multi-agent–optimized generator were run under identical default settings. This provides an initial assessment of cross-domain applicability, while more extensive experiments and task-specific optimizations are left for future work.

For this experiment, results were analyzed from three perspectives. "Overall" refers to the average score over the entire test set. The "Overlap" subset consists of examples whose source tables also appear in the training set but with different subsets of highlighted cells, reflecting performance on tables the model has seen before. The "Non-overlap" subset contains examples whose source tables never appear in the training data, testing generalization to completely unseen tables.

Three automatic evaluation metrics were used: BLEU, PARENT, and BLEURT. BLEU is a traditional n-gram overlap metric that measures similarity between the generated sentence and the reference at the word level(Papineni et al., 2002). PARENT is specifically designed for table- or data-to-text generation, assessing how well the generated text covers the correct table cells and whether it hallucinates content not present in the table (Dhingra et al., 2019). BLEURT is a learned metric based on a finetuned pre-trained language model that leverages semantic representations and human ratings to evaluate adequacy and fluency, and is more tolerant of paraphrases than pure n-gram overlap metrics (Sellam et al., 2020).

The results, shown in Table A11, demonstrate cross-domain applicability of the closed-loop multi-agent plus adaptive reward design framework while restricting modifications to lightweight tool- and lexicon-level adaptations for the ToTTo task.

Table A11: Comparison of the T5 baseline and T5-based MAC-AMP agent framework for the English table-to-text generation task

| Data Subset | Model | BLEU | PARENT | BLEURT |
|---|---|---|---|---|
| Overall | T5 Baseline | 44.6 | 56.0 | 0.179 |
| | T5-based MAC-AMP | 46.2 | 58.0 | 0.208 |
| Overlap | T5 Baseline | 52.6 | 60.7 | 0.311 |
| | T5-based MAC-AMP | 54.1 | 62.4 | 0.338 |
| Non-Overlap | T5 Baseline | 36.8 | 51.4 | 0.051 |
| | T5-based MAC-AMP | 38.5 | 53.7 | 0.083 |

## O  LIMITATIONS AND FUTURE WORK

Our work demonstrates significant advances in AMP drug design and discovery, as well as in LLM-based MAC systems. However, certain limitations remain. First, sensitivity to evaluators and out-of-distribution (OOD) drift persists as our framework is still influenced by the robustness of upstream evaluators and by distributional shifts. The generator can explore sequence space beyond the training distribution, and multi-agent consensus can unintentionally imprint systematic biases onto the reward landscape. Future work will investigate OOD-aware reward specification and adaptively calibrated lexicon weights to enhance robustness. A second limitation concerns closed-loop incentives and exploration diversity. By compiling multi-agent consensus into executable rewards and updating them in a closed loop, multi-objective progress is stabilized, but over long horizons, this approach may favour signals that are easiest to optimize and agree upon, thereby narrowing exploratory diversity. To address this, future work will explore cross-module calibration and diversity-preserving constraints to mitigate incentive drift, preserve exploration breadth, and improve cross-scenario consistency.

## P  USE OF LARGE LANGUAGE MODELS

We acknowledge the use of large language models as a writing assistant in the preparation of this manuscript. Specifically, it was used to help polish the language and writing for flow and clarity. All research ideation, code development and execution, as well as initial drafting and section formatting, were carried out by the authors.

