# OpenReview forum: "MAC-AMP: A Closed-Loop Multi-Agent Collaboration System for Multi-Objective Antimicrobial Peptide Design"
_ICLR.cc/2026/Conference — ICLR 2026 Poster_

### Official Review · Reviewer_yJGN · 2025-10-28

**Soundness:** 3
**Presentation:** 3
**Contribution:** 3
**Rating:** 6
**Confidence:** 4

**Summary:**

MAC-AMP introduces a closed-loop, multi-agent LLM system for multi-objective antimicrobial peptide (AMP) design. The framework integrates three stages: a property prediction module using bioinformatics tools, an AI-simulated peer review system where multiple LLMs collaboratively evaluate peptides along activity, safety, and originality dimensions, and a reinforcement learning refinement stage that converts their structured consensus into executable reward functions for the generator.

**Strengths:**

1. The integration of multi-agent collaboration and reinforcement learning is innovative and potentially impactful for scientific design workflows.
2. The structured peer-review mechanism and log-based audit trail improve transparency relative to typical black-box AMP generators.
3. The modular design could be adapted to other molecule or protein design domains.

**Weaknesses:**

The framework relies on large proprietary models (GPT-5, Perplexity, and Gemini 2.5), which are not easily accessible or affordable for most users. It would be helpful if the authors could provide an estimation of the computational cost for a single complete run, or discuss how the system performs when these agents are replaced with smaller open-source models such as Qwen. Additionally, reporting the total training time compared to a standard non-agent GRPO baseline would help clarify the efficiency and scalability of the proposed approach.

**Questions:**

see weakness

---

> ### Author Response · Authors · 2025-11-26
>
> We sincerely thank you for their thoughtful assessment of our work and for recognizing the novelty and potential impact of MAC-AMP’s multi-agent collaboration, AI-simulated peer review, and reinforcement learning framework. We appreciate the constructive feedback regarding accessibility and computational efficiency and have acknowledged your suggestions below:
>
> **W1 – Computational costs associated with large proprietary models (GPT-5, Perplexity, and Gemini 2.5)**
>
> Thank you for suggesting that we report the computational costs associated with MAC-AMP and its breakdown by large proprietary models. First, to train MAC-AMP for *E. coli* AMP generation, we used 47.61 GPU hours, made 853 API calls, consumed 9106 MB of peak memory, and incurred a total API token cost of $36.56 USD.
>
> To evaluate cost differences associated with Reviewer and Area Chair agents, we performed an additional analysis where we tested replacing the Reviewer and Area Chair agents with Qwen 2.5 7B-instruct and/or LLaMA 3.1 8B, which are locally deployed small models. As shown in Table A7, MAC-AMP achieves the best overall balance across all metrics, excelling in toxicity and AMP likelihood while maintaining competitive antibacterial activity and structural reliability. Variants with higher antibacterial activity or structural reliability incur substantially higher toxicity (2–3×) and lower AMP likelihood. By jointly prioritizing all objectives, MAC-AMP provides the most effective overall performance. Although it has the highest API and computational costs (except peak memory) as seen in Table R1 (and Figure A10 in manuscript), performance is prioritized over efficiency, keeping costs acceptable for the intended use. More details can be found in Appendix Section M.3.
>
> **Table A7: Substitution analyses on the AI-simulated Peer Review Module Agents: property scores of antimicrobial peptides generated by MAC-AMP and its variants. RA AC Llama: all Reviewer agents (RA) and the Area Chair (AC) agent are replaced with Llama 3.1 8B, RA AC Qwen: all RA and the AC agent are replaced with Qwen 2.5 7B-instruct, RA API AC Qwen: only the AC agent was replaced with Qwen, RA Qwen AC API: only the RA are replaced with Qwen**
> | Model          | Antibacterial Activity (↑) | AMP Likelihood (↑) | Toxicity (↓) | Structural Reliability (↑) |
> |----------------|----------------------------|-------------------|-------------|----------------------------|
> | MAC-AMP        | 0.943 ± 0.008             | 0.797 ± 0.012     | 0.154 ± 0.008 | 0.873 ± 0.009             |
> | RA_AC_Llama    | 0.944 ± 0.007             | 0.713 ± 0.012     | 0.201 ± 0.020 | 0.896 ± 0.012             |
> | RA_AC_Qwen     | 0.949 ± 0.009             | 0.683 ± 0.010     | 0.308 ± 0.013 | 0.913 ± 0.016             |
> | RA_API_AC_Qwen | 0.865 ± 0.014             | 0.783 ± 0.011     | 0.206 ± 0.013 | 0.863 ± 0.007             |
> | RA_Qwen_AC_API | 0.835 ± 0.023             | 0.734 ± 0.017     | 0.211 ± 0.030 | 0.854 ± 0.015             |
>
> **Table R1: Substitution analyses on the AI-simulated Peer Review Module Agents: computational costs of antimicrobial peptides generated by MAC-AMP and its variants.**
> | Model Variant        | Total GPU Hours | Peak Memory (MB) | Total API Calls | RL Module API Calls (% of Total) | Reviewer Module API Calls (% of Total) | Total Cost (USD) | OpenAI Cost (% of Total) | Perplexity Cost (% of Total) | GenAI Cost (% of Total) |
> |---------------------|----------------|-----------------|----------------|------------------------|---------------------------------------|-----------------|----------------|-------------------|----------------|
> | Reviewer_AC_llama    | 19.92          | 14757           | 155            | 100.00                 | 0.00                                  | 5.02            | 100.00         | 0.00              | 0.00           |
> | Reviewer_AC_Qwen     | 23.43          | 14219           | 151            | 100.00                 | 0.00                                  | 4.76            | 100.00         | 0.00              | 0.00           |
> | Reviewer_API_AC_Qwen | 31.50          | 14217           | 685            | 22.63                  | 77.37                                 | 26.36           | 75.42          | 8.19              | 16.39          |
> | Reviewer_Qwen_AC_API | 26.54          | 14207           | 334            | 45.81                  | 54.19                                 | 15.01           | 100.00         | 0.00              | 0.00           |
> | MAC-AMP              | 47.61          | 9106            | 853            | 15.12                  | 84.88                                 | 36.56           | 82.69          | 5.61              | 11.71          |
>
> Also, based on your suggestion, for any analyses where we computed API costs, we further broke them down by contribution from GPT-5, Perplexity, and Gemini 2.5.

---

> > ### Author Response · Authors · 2025-11-26
> >
> > **W2 – Computational costs for training**
> >
> > Thank you for suggesting that we report the computational costs associated with MAC-AMP and its comparison to other models. To train MAC-AMP for  *E. coli* AMP generation, we used 47.61 GPU hours, made 853 API calls, consumed 9106 MB of peak memory, and incurred a total API token cost of $36.56 USD. A comparative baseline would be AMPGAN v2, which takes 16 GPU hours.  Although we acknowledge that our model requires more computational power, we believe it is worth the trade-off for generating high-quality AMPs that are designed for multiple objectives.

---

### Official Review · Reviewer_xSXc · 2025-10-29

**Soundness:** 1
**Presentation:** 2
**Contribution:** 2
**Rating:** 2
**Confidence:** 3

**Summary:**

The paper proposes MAC-AMP, a closed-loop multi-agent framework for multi-objective antimicrobial peptide design. It compiles structured “peer review” consensus into PPO-executable rewards to jointly optimize activity, toxicity, and structural reliability, and reports computational improvements over several baselines.
1) The work is difficult to reproduce at review time. Code and data are not publicly available; while the authors commit to releasing them upon acceptance, independent replication and verification cannot be performed during evaluation.
2) Core evaluation and constraints depend on external tools (e.g., ToxinPred 3.0, OmegaFold, Macrel, Foldseek). The reward and selection are tightly coupled to their outputs, yet there is no systematic robustness or bias analysis to assess how these components influence optimization outcomes.
3) The use of OmegaFold pLDDT as a structural reliability proxy for short peptides is insufficiently validated. The paper lacks cross-checks against experimental structures or more rigorous dynamical simulations to establish suitability in the short-peptide regime.
4)The comparison uses a uniform protocol of generating 1,000 candidates and selecting the top-30, which can amplify scorer preferences and ranking bias. The fairness and implications of this selection strategy are not discussed in depth.
5) The paper does not provide rigorous alignment of hyperparameters, training budgets, and screening strategies across baselines, nor does it include sensitivity analyses to quantify how these choices affect reported performance.

**Strengths:**

The paper introduces a novel closed-loop multi-agent framework that compiles structured peer-review consensus into PPO-executable rewards, enabling interpretable and auditable multi-objective optimization for antimicrobial peptide design. Empirically, it reports consistent computational improvements on activity, toxicity, and structural reliability metrics over several baselines, supported by a clearly described system architecture and informative ablation studies.

**Weaknesses:**

1) The work is difficult to reproduce at review time. Code and data are not publicly available; while the authors commit to releasing them upon acceptance, independent replication and verification cannot be performed during evaluation.
2) Core evaluation and constraints depend on external tools (e.g., ToxinPred 3.0, OmegaFold, Macrel, Foldseek). The reward and selection are tightly coupled to their outputs, yet there is no systematic robustness or bias analysis to assess how these components influence optimization outcomes.
3) The use of OmegaFold pLDDT as a structural reliability proxy for short peptides is insufficiently validated. The paper lacks cross-checks against experimental structures or more rigorous dynamical simulations to establish suitability in the short-peptide regime.
4)The comparison uses a uniform protocol of generating 1,000 candidates and selecting the top-30, which can amplify scorer preferences and ranking bias. The fairness and implications of this selection strategy are not discussed in depth.
5) The paper does not provide rigorous alignment of hyperparameters, training budgets, and screening strategies across baselines, nor does it include sensitivity analyses to quantify how these choices affect reported performance.

**Questions:**

1)  Please specify a concrete timeline and scope for releasing code, data, and pretrained models, including licenses, repository URL, and whether an anonymized artifact will be available during rebuttal.
2) Provide an exact, machine-readable environment (Docker/Conda) with pinned versions for all dependencies, especially third-party predictors (e.g., ToxinPred, OmegaFold, Macrel, Foldseek) and their model checkpoints.

---

> ### Author Response · Authors · 2025-11-26
>
> We sincerely thank you for a detailed evaluation of our work. We appreciate your recognition of the novelty of MAC-AMP in translating structured peer-review consensus into PPO-executable rewards, enabling interpretable and auditable multi-objective optimization. We also value your constructive feedback and have addressed your comments and questions below:
>
> **W1 – Code and Data Availability**
>
> Please find all the code and data available at https://github.com/anonymouscat218-creator/anonymous_multi_agent_framework_v1. We are continuing to refine this GitHub, but wanted to provide the first version.
>
> **W2 – Accuracy of Property Predictors**
>
> We understand that the quality of the generated AMPs is constrained by the accuracy of the property predictors. We have summarized how we decided and tested our property predictors below:
>
> **Toxicity:** For MAC-AMP, we use ToxinPred3.0 as the toxicity property prediction model. At the time of ToxinPred 3.0's release [1], it remained one of the top general toxicity prediction models [1]. We tested an “end-to-end” deep learning alternative not tested by ToxinPred 3.0, CAPTP [2] to see whether ToxinPred3.0 is the superior choice for our purposes. We evaluated both ToxinPred 3.0 and CAPTP on an independent dataset originally used to test ToxinPred3.0, ensuring that none of the compounds overlapped with the training sets of either model. As detailed in Table A8, **ToxinPred 3.0 outperforms CAPTP**. More details can be found in Appendix section M.4.
>
> **Table A8. Performance comparison between ToxinPred 3.0 and CAPTP.**
> | **Model**       | **Accuracy** | **Weighted Precision** | **Weighted Recall** | **Weighted F1-score** |
> |-----------------|--------------|------------------------|----------------------|------------------------|
> | ToxinPred 3.0   | 0.828        | 0.849                  | 0.828                | 0.828                  |
> | CAPTP           | 0.770        | 0.787                  | 0.770                | 0.770                  |
>
> **Folding Stability:** We further validated peptide folding stability predicted by OmegaFold via Molecular Dynamics (MD) simulations on a random subset of MAC-AMP–generated peptides using OpenMM (v8.4.0). Backbone Root Mean Square Deviation (RMSD) was calculated to assess structural stability over time relative to the initial structure, providing a general check of OmegaFold’s reliability for guiding structurally robust AMP design. Table A1 shows that the mean RMSD lies around 2-4 Å for the peptides generated by MAC-AMP. This shows that the **peptides designed by MAC-AMP demonstrate general structural stability and thus, using OmegaFold pLDDT as a proxy to guide the generation of AMPs is acceptable.** More details regarding the MD simulation setup can be found in Appendix section I.
>
> **Table A1: Backbone root mean square deviation (RMSD) of molecular dynamics simulations of MAC-AMP-generated anti-Escherichia coli antimicrobial peptides (AMPs) (mean $\pm$ standard deviation)**
> | AMP  | RMSD (Å)    |
> |------|------------|
> | 1    | 2.22 ± 0.79 |
> | 2    | 2.93 ± 0.97 |
> | 3    | 2.16 ± 1.12 |
> | 4    | 3.40 ± 1.85 |
> | 5    | 2.17 ± 0.68 |
> | 6    | 2.66 ± 0.53 |
> | 7    | 4.04 ± 1.61 |
> | 8    | 2.91 ± 0.55 |
> | 9    | 2.66 ± 0.51 |
> | 10   | 2.87 ± 0.72 |
> | 11   | 3.63 ± 0.88 |
> | 12   | 2.85 ± 0.65 |
> | 13   | 3.84 ± 1.29 |
> | 14   | 2.87 ± 0.45 |
> | 15   | 4.96 ± 1.19 |
> | 16   | 2.91 ± 0.99 |
> | 17   | 3.18 ± 1.49 |
> | 18   | 2.25 ± 0.94 |
> | 19   | 1.61 ± 0.38 |
> | 20   | 3.27 ± 0.86 |
> | 21   | 2.17 ± 0.64 |
> | 22   | 1.42 ± 0.54 |
> | 23   | 3.32 ± 0.62 |
> | 24   | 1.73 ± 0.57 |
> | 25   | 1.85 ± 0.65 |
> (more MD simulation results shown in manuscript)
>
> **Minimum Inhibitory Concentration (MIC) Predictions:**  For MAC-AMP, we designed an LLM-based regressor (MAC-AMP MIC Predictor), adapted from BERT AmPEP60, that fine-tunes ProtBERT via transfer learning to map peptide sequences to species-specific MIC values. We compare our designed MIC prediction model to APEX [3], an ensemble of deep-learning networks that uses a peptide-sequence encoder coupled with neural predictors of antimicrobial activity. We evaluated both models on an independent dataset from DBAASP, and as shown in Table A9, **our predictor outperforms APEX**. More details can be found in Appendix Section M.5.
>
> **Table A9: Performance comparison between MAC-AMP MIC Predictor and APEX.**
> | Metric                  | R²     | Pearson r | Spearman ρ |
> |-------------------------|--------|-----------|------------|
> | MAC-AMP MIC Predictor    | **0.572** | **0.758**  | **0.724**   |
> | APEX                     | 0.546  | 0.728     | 0.607      |
>
> References:
>
> [1] Anand Singh Rathore et al., ToxinPred 3.0: An improved method for predicting the toxicity of peptides
>
> [2] Shihu Jiao et al., Integrated convolution and self-attention for improving peptide toxicity prediction
>
> [3] Fangping Wan et al., Deeplearning-enabled antibiotic discovery through molecular de-extinction

---

> ### Author Response · Authors · 2025-11-26
>
> **W3 (Part 1) – Validation of OmegaFold pLDDT as a structural reliability proxy**
>
> Thank you for pointing this out, and we understand the importance of ensuring that using OmegaFold pLDDT values as a structural reliability proxy is accurate.
>
> To address this concern, we performed Molecular Dynamics (MD) simulations on a random subset of MAC-AMP–generated peptides using OpenMM (v8.4.0). Backbone Root Mean Square Deviation (RMSD) was calculated to assess structural stability over time relative to the initial structure, providing a general check of OmegaFold’s reliability for guiding structurally robust AMP design. Table A1 shows that the mean RMSD lies around 2-4 \AA for the peptides generated by MAC-AMP. This shows that the **peptides designed by MAC-AMP demonstrate general structural stability and thus, using OmegaFold pLDDT as a proxy to guide the generation of AMPs is acceptable.** More details regarding the MD simulation setup can be found in Appendix section I.
>
> **Table A1: Backbone root mean square deviation (RMSD) of molecular dynamics simulations of MAC-AMP-generated anti-Escherichia coli antimicrobial peptides (AMPs) (mean $\pm$ standard deviation)**
> | AMP  | RMSD (Å)    |
> |------|------------|
> | 1    | 2.22 ± 0.79 |
> | 2    | 2.93 ± 0.97 |
> | 3    | 2.16 ± 1.12 |
> | 4    | 3.40 ± 1.85 |
> | 5    | 2.17 ± 0.68 |
> | 6    | 2.66 ± 0.53 |
> | 7    | 4.04 ± 1.61 |
> | 8    | 2.91 ± 0.55 |
> | 9    | 2.66 ± 0.51 |
> | 10   | 2.87 ± 0.72 |
> | 11   | 3.63 ± 0.88 |
> | 12   | 2.85 ± 0.65 |
> | 13   | 3.84 ± 1.29 |
> | 14   | 2.87 ± 0.45 |
> | 15   | 4.96 ± 1.19 |
> | 16   | 2.91 ± 0.99 |
> | 17   | 3.18 ± 1.49 |
> | 18   | 2.25 ± 0.94 |
> | 19   | 1.61 ± 0.38 |
> | 20   | 3.27 ± 0.86 |
> | 21   | 2.17 ± 0.64 |
> | 22   | 1.42 ± 0.54 |
> | 23   | 3.32 ± 0.62 |
> | 24   | 1.73 ± 0.57 |
> | 25   | 1.85 ± 0.65 |
> | 26   | 2.50 ± 0.63 |
> | 27   | 3.15 ± 0.71 |
> | 28   | 2.70 ± 0.86 |
> | 29   | 1.80 ± 0.64 |
> | 30   | 1.79 ± 1.04 |
> (more MD simulation results shown in manuscript)
>
> **W3 (Part 2) – Ranking Bias Regarding Comparison of Top 30 Peptides**
>
> Thank you for bringing this to our attention. Our initial motivation behind using the top-30 was to compare the “best-performing” peptides of each model. However, we understand that this may lead to ranking bias as different models may prioritize different features for AMP design. As such, we have decided to perform the physicochemical comparison using 2000 randomly chosen peptides from each model to compare models without this influence.
>
> The results of this analysis are found in Figure A6. Overall, MAC-AMP effectively follows a similar distribution to real-world anti-*E.coli*  AMPs while also optimizing for key properties that help with antimicrobial activity such as:
> - Amino acid composition is enriched in K/R and L/I residues, reflecting cationic, amphipathic α-helices
> - Peptides peak at a global charge of +5–7
> - Hydrophobicity (~0.5) balances solubility and membrane insertion
> - Predicted helix fractions (~0.45–0.5) and lengths (17–20 aa) support stable helices
> - Eisenberg hydrophobic moments (~0.55 median) generate well-defined amphipathic faces
>
> Together, these results indicate that **MAC-AMP maintains natural AMP characteristics while enhancing features favourable for antimicrobial activity.** This analysis has been added to the revised manuscript in Appendix K.

---

> ### Author Response · Authors · 2025-11-26
>
> **W4 – Lack of hyperparameters, training budgets and screening strategies across baselines and sensitivity analyses**
>
> We agree that fair comparison across models is important. All baseline methods we evaluated are publicly available models with downloadable weights, and we used each according to the inference and configuration settings specified in their original papers. Because these baselines differ substantially in architecture, each is optimized for its own training regime; forcing them into a single, unified set of hyperparameters would produce non-standard, suboptimal variants of the original models. For this reason, we follow standard practice and evaluate each baseline using its published configuration, while standardizing only the task-relevant factor of maximum peptide length (set to 32 for all models). To enable fair comparison across our model and baseline models, we applied a consistent screening strategy: sequences with predicted toxicity > 0.3 were discarded, followed by filtering out sequences with structural reliability < 0.6 and AMP likelihood < 0.7. If a baseline generated fewer than 30 sequences out of 1,000 that passed a given criterion, that criterion was omitted. The remaining sequences were then ranked by predicted antibacterial activity, and the top 90 sequences were selected for comparison.
>
> For MAC-AMP, we performed extensive ablation studies and substitution analyses, including tests of module contributions, reviewer-agent configurations, RL reward and optimization mechanisms, swapped property-prediction models, and hyperparameter/computational-scaling effects. Our experiments are summarized in Table R6. They are all further detailed with results in Appendix Sections L & M.
>
> **Table R6: Overview of Sensitivity Analyses Performed and Objectives**
> | Module / Component            | Sensitivity Analysis                                                                | Purpose                                                                                                 |
> |-------------------------------|---------------------------------------------------------------------------------|---------------------------------------------------------------------------------------------------------|
> | Property Prediction Module    | Remove individual objectives: Toxicity (Va), Structural Reliability (Vb), AMP Likelihood (Sb), or combinations | Assess contribution of each objective to performance and computational costs in multi-objective optimization |
> | AI-simulated Peer Review Module | Remove one or more reviewer agents (GPT-5, Perplexity, Gemini 2.5)             | Determine how reviewers balance potency, safety, and structural reliability, and whether all 3 agents are necessary for optimal performance |
> | RL Refinement Module          | Remove RL Reward Decision agent and/or Adaptive Optimization; replace with human-designed RL | Evaluate the role of adaptive RL and its components in balanced multi-objective optimization on model performance |
> | RL Training Epochs            | Test 10, 15 (default), and 20 epochs                                            | Test scalability of RL training epochs by analyzing computational cost differences                      |
> | Maximum Peptide Length        | Test 20, 26, 32 (default), and 39 maximum peptide length                         | Test scalability of peptide length by analyzing computational cost differences                            |
> | Reviewer Agents               | Replace GPT-5 / Perplexity / Gemini with smaller local models (Llama 3.1 8B, Qwen 2.5 7B) in various configurations | Evaluate the contribution of reviewer models to computational cost and performance                       |
> | Toxicity Predictor            | Replace ToxinPred 3.0 with CAPTP                                                | Validate the choice of toxicity prediction model                                                         |
> | MIC Predictor                 | Replace MAC-AMP MIC predictor with APEX                                         | Validate the choice of MIC prediction model                                                             |
> | Lexicon Weights               | Offset expert-assigned lexicon weights in the AI-simulated peer review map by ±0.1 | Assess robustness of peer-review lexicon weight initialization on model performance                       |

---

> ### Author Response · Authors · 2025-11-26
>
> **Q1 - Release of code, data, and pretrained models, including licenses and repository URL**
>
> Please find all the code and data available at https://github.com/anonymouscat218-creator/anonymous_multi_agent_framework_v1. We are continuing to refine this GitHub, but wanted to provide the first version.
>
> The requested components are provided in the following locations:
> - Data: All training data, organized by bacterial species for folder naming, can be found in the `data/` folder.
> - Pretrained models: Third-party pretrained predictors must be downloaded separately; instructions are provided in Section 3 of the README.
> - Licensing information: Complete licensing details are available in Section 4 of the README.
>
> **Q2 - Need for Machine-readable environment**
>
> Please find all the code and data available at https://github.com/anonymouscat218-creator/anonymous_multi_agent_framework_v1. We are continuing to refine this GitHub, but wanted to provide the first version.
>
> The Conda environment with exact, pinned dependencies is provided in conda_environment.yml. Third-party predictors must be installed independently; installation instructions are detailed in Section 3 of the README. Note that we cannot host the model checkpoints on GitHub, and we did not retrain any of them. Our framework simply uses the specified versions as provided by the original tools.

---

### Official Review · Reviewer_yaxY · 2025-10-31

**Soundness:** 3
**Presentation:** 3
**Contribution:** 2
**Rating:** 4
**Confidence:** 3

**Summary:**

The paper introduces MAC-AMP, a closed-loop multi-agent collaboration framework for multi-objective antimicrobial peptide design, tackling the complex trade-offs between activity, toxicity, and stability. The system integrates LLM-based agents (e.g., GPT-5, Gemini 2.5, Perplexity) within an AI-driven peer-review loop that evaluates candidate peptides along multiple criteria and transforms the structured consensus into executable reinforcement learning reward signals. A biomedical–computer science dual-agent team iteratively refines these rewards, guiding a PPO-based generator to optimize AMP sequences under continuous feedback—achieving interpretable and auditable improvements. Across E. coli, S. aureus, and P. aeruginosa, MAC-AMP consistently surpasses baselines such as AMP-Designer, BroadAMP-GPT, PepGAN, and Diff-AMP in antibacterial efficacy, toxicity regulation, and structural robustness, while preserving AMP-like properties and strong cross-species generalization. Collectively, MAC-AMP establishes a generalizable and interpretable paradigm for autonomous molecular design.

**Strengths:**

The paper introduces a closed-loop multi-agent collaboration framework that transforms AMP design into an autonomous, explainable reinforcement learning process. Its integration of AI-simulated peer review to generate executable reward signals is both novel and technically elegant. It is clearly written, with transparent modular design and reproducible details. Overall, the paper is significant, offering a transferable paradigm for multi-objective molecular design and broader autonomous scientific discovery.

**Weaknesses:**

While the framework is conceptually strong, several areas could be improved.
1. The novelty is quite limited. Using existing PPO and peptide generation model (Wang et al. 2025)
2. Experimental validation is limited to in silico analyses—no wet-lab or biophysical confirmation is provided to verify that the generated AMPs exhibit real-world antimicrobial activity, which would strengthen the biological significance.
3. Although the AI-simulated peer review concept is creative, the paper could include quantitative ablations demonstrating how each reviewer agent or the Area Chair aggregation contributes to performance, beyond qualitative explanation.
4. The computational cost and scalability of running multi-agent deliberation and RL loops are not clearly analyzed, raising concerns about feasibility for large-scale or real-time applications.
5. While the framework claims cross-domain generalizability, no transfer experiments to non-AMP domains are shown, leaving this claim largely theoretical. Addressing these limitations would substantially enhance the rigor and impact of the work.

**Questions:**

See weakness.

---

> ### Author Response · Authors · 2025-11-26
>
> We sincerely thank you for a thoughtful evaluation of our work. We greatly appreciate your recognition of the novelty of MAC-AMP, particularly the autonomous multi-agent collaboration and AI-simulated peer review that generates executable reward signals. Your insights regarding the limitations of our paper were extremely valuable. Please find our responses to your comments below:
>
> **W1 – Limitations of Novelty**
>
> We acknowledge using a standard PPO and an AMP Designer–family head. The novelty of our framework lies in the **closed-loop multi-agent review-to-reward architecture, which compiles structured committee consensus into executable RL signals**, featuring stage-wise adaptation and full auditability. We intentionally reused established components to control for confounds, demonstrating that the **framework itself drives gains** via backbone comparisons and ablations reported in the paper.
>
> A key innovation is its **modular “plug-and-play” design: the generator can be replaced or adapted to new domains without altering the surrounding system**, while agents, ontology-guided review, and adaptive reward modules remain unchanged, enabling flexible deployment. **Stepwise explainability through transparent logs, replay traces, and consensus-aware decision tracking mitigates typical “black-box” limitations** and allows precise error or bias correction. The framework also **natively supports multi-objective AMP design**, balancing antibacterial activity, stability, and safety to produce robust, application-ready peptides.
>
> For clarity, we do not claim novelty in PPO or the sequence head. **Our novel contributions include:** (1) a system-level integration linking multi-agent consensus to executable rewards in a closed loop, (2) a modular “plug-and-play” architecture enabling adaptation to new domains without modifying the surrounding system, (3) stepwise explainability to address the “black-box” limitations of most ML models, and (4) native support for multi-objective AMP design, balancing activity, stability, and safety.
>
> **W2 – Lack of Experimental Validation**
>
> We acknowledge that wet-lab validation would further strengthen our results, but as a computational lab, we lack the resources to perform experiments. Given the AI-focused scope of this conference, we provide extensive in silico analyses and additional validations of the generated peptides. Some of our in silico analyses are detailed below:
>
> In Appendix section K, 2000 random AMPs generated to target *E. coli* by MAC-AMP were compared to the training *E. coli* dataset (real-world AMPs) and other baseline models in regards to various physicochemical properties. Results, shown in Figure A6, demonstrate that MAC-AMP faithfully recapitulates the canonical features of natural AMPs. This goes to show that **our designed peptides follow experimentally validated activity peptides.**
>
> We also performed molecular dynamics (MD) simulations on a subset of randomized peptides generated by MAC-AMP to assess general structural stability using OpenMM (v8.4.0). Backbone Root Mean Square Deviation (RMSD) was calculated to assess structural stability over time relative to the initial structure, providing a general check of OmegaFold’s reliability for guiding structurally robust AMP design. Table A1 shows that the mean RMSD lies around 2-4 Å for the peptides generated by MAC-AMP. This shows that the **peptides designed by MAC-AMP demonstrate general structural stability and thus, using OmegaFold pLDDT as a proxy to guide the generation of AMPs is acceptable.** More details regarding the MD simulation setup can be found in Appendix Section I.
>
> **Table A1: Backbone root mean square deviation (RMSD) of molecular dynamics simulations of MAC-AMP-generated anti-Escherichia coli antimicrobial peptides (AMPs) (mean $\pm$ standard deviation)**
> | AMP  | RMSD (Å)    |
> |------|------------|
> | 1    | 2.22 ± 0.79 |
> | 2    | 2.93 ± 0.97 |
> | 3    | 2.16 ± 1.12 |
> | 4    | 3.40 ± 1.85 |
> | 5    | 2.17 ± 0.68 |
> | 6    | 2.66 ± 0.53 |
> | 7    | 4.04 ± 1.61 |
> | 8    | 2.91 ± 0.55 |
> | 9    | 2.66 ± 0.51 |
> | 10   | 2.87 ± 0.72 |
> | 11   | 3.63 ± 0.88 |
> | 12   | 2.85 ± 0.65 |
> | 13   | 3.84 ± 1.29 |
> | 14   | 2.87 ± 0.45 |
> | 15   | 4.96 ± 1.19 |
> | 16   | 2.91 ± 0.99 |
> | 17   | 3.18 ± 1.49 |
> | 18   | 2.25 ± 0.94 |
> | 19   | 1.61 ± 0.38 |
> | 20   | 3.27 ± 0.86 |
> (more MD simulation results shown in manuscript)
>
> To further validate MAC-AMP peptides in silico, we analyzed the top 90 generated sequences against  *E. coli* for motifs associated with antibacterial activity. 8 peptides contained CRAC or CARC motifs, which may disrupt bacterial membranes, and 2 peptides are proline-rich antimicrobial peptides (PR-AMPs) with arginine-rich regions and repeating Pro-Arg motifs, characteristic of antibacterial activity. These findings provide **mechanistic evidence supporting MAC-AMP–designed peptides.** More details can be found in Appendix section H.

---

> ### Author Response · Authors · 2025-11-26
>
> **W3 – Sensitivity of MAC-AMP to Reviewer Agent LLMs**
>
> Thank you for pointing out the importance of ensuring the correct choice in the number and specific Reviewer Agent LLMs in our model.  To ensure that we are not unnecessarily using extra Reviewer agents, we performed ablation studies where we specifically removed each of the Reviewer agents, combinations of them, and all of them to see whether fewer agents can achieve similar performance. As seen in Table A5, the **removal of any reviewer agents resulted in decreased model performance, justifying the requirement for all 3 reviewer agents**. This is detailed in Appendix section L.2.
>
> **Table A5: Ablation studies of the AI-simulated Peer Review module: property scores of antimicrobial peptides generated by MAC-AMP and its variants. RA1 = Reviewer agent 1 (GPT-5), RA2 = Reviewer agent 2 (Perplexity), RA3 = Reviewer agent 3 (Gemini 2.5)**
> | Model            | **Antibacterial Activity (↑)** | **AMP Likelihood (↑)** | **Toxicity (↓)**     | **Structural Reliability (↑)** |
> |------------------|--------------------------------|-------------------------|------------------------|---------------------------------|
> | MAC-AMP          | 0.943 ± 0.008                  | 0.797 ± 0.012           | 0.154 ± 0.008          | 0.873 ± 0.009                   |
> | w/o RA3          | 0.857 ± 0.016                  | 0.840 ± 0.008           | 0.133 ± 0.023          | 0.731 ± 0.017                   |
> | w/o RA2          | 0.884 ± 0.015                  | 0.780 ± 0.008           | 0.164 ± 0.023          | 0.734 ± 0.021                   |
> | w/o RA1          | 0.953 ± 0.010                  | 0.742 ± 0.016           | 0.181 ± 0.019          | 0.783 ± 0.017                   |
> | w/o RA2 + RA3    | 0.870 ± 0.013                  | 0.806 ± 0.006           | 0.130 ± 0.015          | 0.740 ± 0.011                   |
> | w/o RA1 + RA3    | 0.979 ± 0.010                  | 0.700 ± 0.019           | 0.185 ± 0.019          | 0.758 ± 0.015                   |
> | w/o RA1 + RA2    | 0.938 ± 0.011                  | 0.795 ± 0.008           | 0.195 ± 0.013          | 0.862 ± 0.013                   |
> | w/o All Review   | 0.825 ± 0.016                  | 0.627 ± 0.014           | 0.448 ± 0.051          | 0.831 ± 0.012                   |
>
>
> To ensure that we chose optimal LLM APIs, we performed an additional analysis where we tested replacing the Reviewer and Area Chair agents with Qwen 2.5 7B-instruct and/or LLaMA 3.1 8B, which are locally deployed small models. As shown in Table A7, MAC-AMP achieves the best overall balance across all metrics, excelling in toxicity and AMP likelihood while maintaining competitive antibacterial activity and structural reliability. Variants with higher antibacterial activity or structural reliability incur substantially higher toxicity (2–3×) and lower AMP likelihood. By jointly prioritizing all objectives, MAC-AMP provides the most effective overall performance. Although it has the highest API and computational costs (except peak memory) as seen in  Figure A10 in the manuscript, performance is prioritized over efficiency, keeping costs acceptable for the intended use. More details can be found in Appendix Section M.3.
>
> **Table A7: Substitution analyses on the AI-simulated Peer Review Module Agents: property scores of antimicrobial peptides generated by MAC-AMP and its variants. RA\_AC\_Llama: all Reviewer agents (RA) and the Area Chair (AC) agent are replaced with Llama 3.1 8B, RA\_AC\_Qwen: all RA and the AC agent are replaced with Qwen 2.5 7B-instruct, RA\_API\_AC\_Qwen: only the AC agent was replaced with Qwen, RA\_Qwen\_AC\_API: only the RA are replaced with Qwen.**
> | **Model**          | **Antibacterial Activity (↑)** | **AMP Likelihood (↑)** | **Toxicity (↓)**      | **Structural Reliability (↑)** |
> |--------------------|--------------------------------|--------------------------|-------------------------|---------------------------------|
> | MAC-AMP            | 0.943 ± 0.008                  | 0.797 ± 0.012            | 0.154 ± 0.008           | 0.873 ± 0.009                   |
> | RA_AC_Llama        | 0.944 ± 0.007                  | 0.713 ± 0.012            | 0.201 ± 0.020           | 0.896 ± 0.012                   |
> | RA_AC_Qwen         | 0.949 ± 0.009                  | 0.683 ± 0.010            | 0.308 ± 0.013           | 0.913 ± 0.016                   |
> | RA_API_AC_Qwen     | 0.865 ± 0.014                  | 0.783 ± 0.011            | 0.206 ± 0.013           | 0.863 ± 0.007                   |
> | RA_Qwen_AC_API     | 0.835 ± 0.023                  | 0.734 ± 0.017            | 0.211 ± 0.030           | 0.854 ± 0.015                   |

---

> ### Author Response · Authors · 2025-11-26
>
> **W4 – Computational Costs and Scalability with Multi-Agent Deliberation and RL Loops**
>
> Thank you for these suggestions. First, to train MAC-AMP for  *E. coli* AMP generation, we used 47.61 GPU hours, made 853 API calls, consumed 9106 MB of peak memory, and incurred a total API token cost of $36.56 USD.
>
> To dissect scalability in multi-agent deliberation, we decoupled the size of the underlying generative model from the RL training cost. Large models are first adapted via a frozen soft prompt and then distilled into a lightweight generator with fixed architecture. During RL, only this small generator is updated, while the backbone remains frozen. This allows the generator to inherit the expressive power of the large model without increasing training cost, ensuring that multi-agent deliberation remains computationally efficient as the backbone scales. The results of this approach are detailed in Table R4, where **we observed that replacing the backbone with Llama models of 1B and 8B parameters resulted in essentially unchanged runtime, demonstrating that our multi-agent framework scales effectively with model size**.
>
> **Table R4:  Substitution analyses on the Number of Generative Model Parameters: computational costs of antimicrobial peptides generated by MAC-AMP and its variants**
> | Model Variant                  | Total GPU Hours | Peak Memory (MB) | Total API Calls | RL Module API Calls (% of Total) | Reviewer Module API Calls (% of Total) | Total Cost (USD) | OpenAI Cost (% of Total) | Perplexity Cost (% of Total) | GenAI Cost (% of Total) |
> |--------------------------------|----------------|-----------------|----------------|---------------------------------|---------------------------------------|-----------------|-------------------------|-----------------------------|-------------------------|
> | generation_module_llama_1b     | 47.15          | 6292            | 851            | 14.92                           | 85.08                                 | 36.62           | 83.32                   | 5.35                        | 11.33                   |
> | generation_module_llama_8b     | 47.24          | 6624            | 885            | 18.19                           | 81.81                                 | 38.07           | 83.87                   | 5.15                        | 10.98                   |
> | MAC-AMP_generation_size_85M    | 47.61          | 9106            | 853            | 7.24                            | 84.88                                 | 36.56           | 82.69                   | 5.61                        | 11.71                   |
>
>
> To further understand the RL computational cost, we evaluated 10, 15 (default), and 20 training epochs for the RL Refinement module. Results are shown in **Table R5** (Figure A8 in manuscript), and demonstrate that increasing from 10 to 15 epochs only adds **\$0.46 USD**, while 20 epochs increases the cost by **\$5.62 USD**. GPU hours rise modestly with more epochs, whereas peak memory remains largely unchanged.
>
> **Table R5: Substitution analyses on the Number of Training Epochs for the RL Refinement module: computational costs of antimicrobial peptides generated by MAC-AMP and its variants**
> | Model     | Total GPU Hours | Peak Memory (MB) | Total API Calls | RL Module API Calls (% of Total) | Reviewer Module API Calls (% of Total) | Total Cost (USD) | OpenAI Cost (% of Total) | Perplexity Cost (% of Total) | GenAI Cost (% of Total) |
> |------------------|----------------|-----------------|----------------|------------------------|-------------------------------|-----------------|----------------|-------------------|----------------|
> | 10_epoch         | 43.86          | 9101            | 795            | 16.48                  | 83.52                         | 36.10           | 82.91          | 5.40              | 11.69          |
> | MAC_AMP_15_epoch | 47.61          | 9106            | 853            | 15.12                  | 84.88                         | 36.56           | 82.69          | 5.61              | 11.71          |
> | 20_epoch         | 52.38          | 9137            | 917            | 14.50                  | 85.50                         | 42.18           | 82.41          | 5.55              | 12.04          |

---

> ### Author Response · Authors · 2025-11-26
>
> **W5 – Domain Transferability Proof**
>
> One of MAC-AMP’s biggest novelties is its ability for cross-domain transferability. To test its potential beyond AMP design, we performed a small domain transfer experiment for an English table-to-text generation task where a given a table and a set of highlighted table cells, the model will produce a one-sentence description. To test this task, the ToTTo dataset was used, an open-domain English table-to-text dataset with over 120,000 training examples of Wikipedia tables and a set of highlighted table cells to produce a one-sentence description.
>
> In transferring MAC-AMP to the ToTTo table-to-text benchmark, the multi-agent architecture, logging scheme, and PPO training code remain unchanged, with only minimal domain-specific substitutions. The Property Prediction module now uses official ToTTo scripts to compute PARENT and BLEU scores, linearly normalized as $S_a$ and $S_b$ for the RL pipeline. Additional statistics (table coverage, proportion of unsupported tokens, and length ratio) are exposed to the reviewer panel as V-style auxiliary evidence.
>
> The reviewer module still evaluates four dimensions (Eff / Safe / DevStruct / Orig) with redefined semantics for text generation:
> - Eff: task effectiveness and content adequacy (coverage of highlighted cells, correctness of facts)
> - Safe: factual safety (absence of hallucinations or contradictions relative to the table)
> - DevStruct: linguistic quality and structural coherence (grammar, fluency, Wikipedia-like discourse)
> - Orig: stylistic diversity and faithful paraphrasing
>
> Weighted lexicons are reconstructed for each dimension, while the original tagging and scoring mechanisms remain. In the RL Refinement and generation modules, the peptide generator is replaced by a T5-small encoder-decoder model. Reward-design and alignment agents are updated to reference PARENT/ BLEU and the redefined four dimensions, emitting TorchScript reward functions $F(S_a, S_b, S_c) \in [0,1]$, which are consumed by the same PPO strategy to update the generator.
>
> Due to time limitations, only a minimal evaluation was conducted in which both the T5-based baseline and the multi-agent–optimized generator was run under identical default settings. For this preliminary study, a randomly sampled subset of 30,000 training examples (approximately 25\% of the full dataset of 120,761 sequences) was used. This provides an initial assessment of cross-domain applicability, while more extensive experiments and task-specific optimizations are left for future work. The results are shown in Table A11. This transfer experiment demonstrates cross-domain applicability of the closed-loop multi-agent plus adaptive reward design framework while restricting modifications to lightweight tool- and lexicon-level adaptations for the ToTTo task.
>
> For this experiment, results were analyzed from three perspectives. “Overall” refers to the average score over the entire test set. The “Overlap” subset consists of examples whose source tables also appear in the training set but with different subsets of highlighted cells, reflecting performance on tables the model has seen before. The “Non-overlap” subset contains examples whose source tables never appear in the training data, testing generalization to completely unseen tables.
>
> The results, shown in Table A11, **demonstrate cross-domain applicability of the closed-loop multi-agent plus adaptive reward design framework while restricting modifications to lightweight tool- and lexicon-level adaptations for the ToTTo task.**
>
> **Table A11: Comparison of T5 Baseline and T5-based MAC-AMP Agent Framework for English table-to-text generation task.**
> | **Subset**   | **Model**           | **BLEU** | **PARENT** | **BLEURT** |
> |--------------|----------------------|----------|------------|------------|
> | Overall      | T5 Baseline          | 44.6     | 56.0       | 0.179      |
> |              | T5-based MAC-AMP     | 46.2     | 58.0       | 0.208      |
> | Overlap      | T5 Baseline          | 52.6     | 60.7       | 0.311      |
> |              | T5-based MAC-AMP     | 54.1     | 62.4       | 0.338      |
> | Non-Overlap  | T5 Baseline          | 36.8     | 51.4       | 0.051      |
> |              | T5-based MAC-AMP     | 38.5     | 53.7       | 0.083      |
>
> This analysis has been added to the revised manuscript in Appendix N.

---

### Official Review · Reviewer_vtM8 · 2025-10-31

**Soundness:** 3
**Presentation:** 3
**Contribution:** 3
**Rating:** 6
**Confidence:** 2

**Summary:**

The paper introduces MAC-AMP, a closed-loop multi-agent collaboration (MAC) system for multi-objective antimicrobial peptide (AMP) design. The system integrates property prediction, AI-simulated peer review, reinforcement learning (RL) refinement, and peptide generation modules in a fully autonomous loop. A key novelty is the conversion of multi-agent textual consensus into executable reward signals, allowing reinforcement-based optimization without manual prompt tuning. Experiments on five bacterial targets show that MAC-AMP outperforms existing generative models in antibacterial activity, toxicity reduction, and structural reliability. Ablation studies confirm that both the AI-simulated peer review and RL refinement modules contribute critically to balanced multi-objective optimization.

**Strengths:**

- The paper introduces a credible pipeline that operationalizes “multi-agent collaboration” beyond conversational coordination, producing quantitative, auditable training signals.
- Transparent logging and role-based agent structure allow reproducibility and traceability uncommon in LLM-based systems.
- The system’s iterative RL refinement aligns multi-agent consensus with executable objectives, avoiding reward hacking typical in static scoring systems.
- Results across multiple bacterial targets and comparisons to several baselines demonstrate consistent gains in activity and safety metrics.
- Includes ablation analyses, sequence-level property comparisons, and visualization of learned chemical space.
- The framework is described as transferable to other molecule or material design problems with minimal adaptation.

**Weaknesses:**

- The quality of generated AMPs is constrained by the accuracy of property predictors (e.g., ToxinPred 3.0, OmegaFold). The paper acknowledges this but does not quantify its impact.
- While broad-spectrum testing is reported, the reasoning behind cross-species generalization (beyond physicochemical similarity) could be more rigorously analyzed.
- Although individual module ablations are provided, it remains unclear whether a simpler reward design or fewer agents would achieve similar performance.

**Questions:**

1. How sensitive is MAC-AMP’s performance to the specific LLMs used as reviewer agents (GPT-5, Gemini 2.5, Perplexity)? Would substituting smaller open models maintain performance trends?

2. The AI-simulated peer review uses a lexicon-tagging mechanism, how were the weights and tag mappings validated for interpretability or bias?

3. Does the system ever experience feedback collapse or overfitting to its own reviewers’ biases after multiple closed-loop iterations?

4. How is novelty quantitatively ensured beyond low Foldseek similarity, are there checks for redundancy or motif-level overlap with training data?

5. Could the authors provide empirical measures of reward variance or entropy across RL stages to confirm that adaptive optimization prevents reward hacking?

6. What are the computational costs (GPU hours, number of agent calls per iteration) and how do they scale with peptide length or number of objectives?

---

> ### Author Response · Authors · 2025-11-26
>
> We sincerely thank you for a thoughtful and thorough assessment of our work. We are grateful for your recognition of the novelty and rigor of MAC-AMP, particularly the operationalization of multi-agent collaboration into a fully autonomous loop with executable reward signals. We also appreciate your constructive observations and insights, which have guided us to clarify these points in the manuscript and better contextualize our framework. We have detailed our responses to your questions and comments below:
>
> **W1 – Accuracy of Property Predictors**
>
> We understand that the quality of the generated AMPs is constrained by the accuracy of the property predictors. To ensure the accuracy and the best choice for our property predictors, we have added the following analyses:
>
> **Toxicity:** For MAC-AMP, we use ToxinPred3.0 as the toxicity property prediction model. At the time of ToxinPred 3.0's release [1], it remained one of the top general toxicity prediction models [1]. We tested an “end-to-end” deep learning alternative not tested by ToxinPred 3.0, CAPTP [2] to see whether ToxinPred3.0 is the superior choice for our purposes. We evaluated both ToxinPred 3.0 and CAPTP on an independent dataset originally used to test ToxinPred3.0, ensuring that none of the compounds overlapped with the training sets of either model. As detailed in Table A8, **ToxinPred 3.0 outperforms CAPTP**. More details can be found in Appendix section M.4.
>
> **Table A8. Performance comparison between ToxinPred 3.0 and CAPTP**
> | **Model**       | **Accuracy** | **Weighted Precision** | **Weighted Recall** | **Weighted F1-score** |
> |-----------------|--------------|------------------------|----------------------|------------------------|
> | ToxinPred 3.0   | 0.828        | 0.849                  | 0.828                | 0.828                  |
> | CAPTP           | 0.770        | 0.787                  | 0.770                | 0.770                  |
>
> **Folding Stability:** We further validated peptide folding stability predicted by OmegaFold via Molecular Dynamics (MD) simulations on a random subset of MAC-AMP–generated peptides using OpenMM (v8.4.0). Backbone Root Mean Square Deviation (RMSD) was calculated to assess structural stability over time relative to the initial structure, providing a general check of OmegaFold’s reliability for guiding structurally robust AMP design. Table A1 shows that the mean RMSD lies around 2-4 Å for the peptides generated by MAC-AMP. This shows that the **peptides designed by MAC-AMP demonstrate general structural stability and thus, using OmegaFold pLDDT as a proxy to guide the generation of AMPs is acceptable.** More details regarding the MD simulation setup can be found in Appendix section I.
>
> **Table A1: Backbone root mean square deviation (RMSD) of molecular dynamics simulations of MAC-AMP-generated anti-Escherichia coli antimicrobial peptides (AMPs) (mean $\pm$ standard deviation)**
> | AMP  | RMSD (Å)    |
> |------|------------|
> | 1    | 2.22 ± 0.79 |
> | 2    | 2.93 ± 0.97 |
> | 3    | 2.16 ± 1.12 |
> | 4    | 3.40 ± 1.85 |
> | 5    | 2.17 ± 0.68 |
> | 6    | 2.66 ± 0.53 |
> | 7    | 4.04 ± 1.61 |
> | 8    | 2.91 ± 0.55 |
> | 9    | 2.66 ± 0.51 |
> | 10   | 2.87 ± 0.72 |
> | 11   | 3.63 ± 0.88 |
> | 12   | 2.85 ± 0.65 |
> | 13   | 3.84 ± 1.29 |
> | 14   | 2.87 ± 0.45 |
> | 15   | 4.96 ± 1.19 |
> | 16   | 2.91 ± 0.99 |
> | 17   | 3.18 ± 1.49 |
> | 18   | 2.25 ± 0.94 |
> | 19   | 1.61 ± 0.38 |
> | 20   | 3.27 ± 0.86 |
> | 21   | 2.17 ± 0.64 |
> | 22   | 1.42 ± 0.54 |
> | 23   | 3.32 ± 0.62 |
> | 24   | 1.73 ± 0.57 |
> | 25   | 1.85 ± 0.65 |
> (more MD simulation results shown in manuscript)
>
> **Minimum Inhibitory Concentration (MIC) Predictions:**  For MAC-AMP, we designed an LLM-based regressor (MAC-AMP MIC Predictor), adapted from BERT AmPEP60, that fine-tunes ProtBERT via transfer learning to map peptide sequences to species-specific MIC values. We compare our designed MIC prediction model to APEX [3], an ensemble of deep-learning networks that uses a peptide-sequence encoder coupled with neural predictors of antimicrobial activity. We evaluated both models on an independent dataset from DBAASP, and as shown in Table A9, **our predictor outperforms APEX**. More details can be found in Appendix Section M.5.
>
> **Table A9: Performance comparison between MAC-AMP MIC Predictor and APEX**
> | Metric                  | R²     | Pearson r | Spearman ρ |
> |-------------------------|--------|-----------|------------|
> | MAC-AMP MIC Predictor    | **0.572** | **0.758**  | **0.724**   |
> | APEX                     | 0.546  | 0.728     | 0.607      |
>
> References:
>
> [1] Anand Singh Rathore et al., ToxinPred 3.0: An improved method for predicting the toxicity of peptides
>
> [2] Shihu Jiao et al., Integrated convolution and self-attention for improving peptide toxicity prediction
>
> [3] Fangping Wan et al., Deeplearning-enabled antibiotic discovery through molecular de-extinction

---

> ### Author Response · Authors · 2025-11-26
>
> **W2 – Analysis of Cross-Species Generalization**
>
> We thank the reviewer for suggesting further analysis of the cross-species generalization of our generated peptides. In response, we (1) externally validated the broad-spectrum activity predicted by MAC-AMP and (2) performed motif analysis across the broad-spectrum generated peptides.
>
> To validate the broad-spectrum activity predicted by MAC-AMP, the 90 generated AMPs were evaluated using the external MIC predictor APEX [1] against multiple ESKAPE strains: *Acinetobacter baumannii* (ATCC 19606), *Klebsiella pneumoniae* (ATCC 13883), *Pseudomonas aeruginosa* (PA01 and PA14), *Staphylococcus aureus* (ATCC 12600 and BAA-1556), and vancomycin-resistant *Enterococcus faecium* (ATCC 700221). First, we see that **all generated peptides exhibit low MIC values against at least one *E. coli* strain.**
>
> Using a MIC cutoff of ≤ 128 μmol·L⁻¹, APEX broad-spectrum testing results were binarized into active versus inactive per strain. Of the top 90 MAC-AMP–generated anti-*E. coli* peptides, 36 (40%) were broad-spectrum, defined as active against at least one strain of each of the five additional ESKAPE pathogens.
>
> To further understand potential mechanisms behind the broad-spectrum activity of these peptides, motif analysis was performed using MEME and FIMO, revealing two notable conserved motifs: KFLKGA and WLLGKW. KFLKGA exhibits an alternating pattern of cationic (K) and hydrophobic (F, L, A) residues, characteristic of amphipathic antimicrobial peptides. Similarly, WLLGKW features a central lysine flanked by hydrophobic tryptophan and leucine residues, with the tryptophans likely enhancing membrane interaction. Both motifs are consistent with the canonical two-step AMP mechanism: electrostatic binding via cationic residues to bacterial membranes, followed by hydrophobic insertion that disrupts the lipid bilayer. **These patterns may underlie the broad-spectrum activity observed in the broad-spectrum generated peptides.**
> The details of this analysis are found in Appendix section F.
>
> Reference:
> [1] Fangping Wan et al., Deep learning-enabled antibiotic discovery through molecular de-extinction.

---

> ### Author Response · Authors · 2025-11-26
>
> **W3 – Validation of Reward Design and Agent Number**
>
> Thank you for highlighting the importance of ensuring that our model framework is not unnecessarily complex. To validate that our reward design methodology and number of agents are appropriate, we conducted the following experiments:
>
> Our reward design is fully automated and decided by the agents in the RL Refinement Module to prioritize performance, so the actual reward is already optimized on the fly. To assess whether the reward design could be simplified, ablation studies evaluated the impact of removing the RL Decision Agent, Adaptive Optimization, both, or replacing them with a human-designed RL procedure. As seen in Table A6, **removing any component of the RL Refinement module results in decreased performance** and thus, justifies our reward design process. More details can be found in Appendix section L.3.
>
> **Table A6: Ablation studies of the Reinforcement Learning Refinement module: property scores of antimicrobial peptides generated by MAC-AMP and its variants**
>
> | Model                                      | Antibacterial Activity (↑) | AMP Likelihood (↑) | Toxicity (↓)      | Structural Reliability (↑) |
> |-------------------------------------------|----------------------------|-------------------|------------------|----------------------------|
> | MAC-AMP                                    | **0.943 ± 0.008**         | **0.797 ± 0.012** | **0.154 ± 0.008** | 0.873 ± 0.009             |
> | Without RL Decision Agent                  | 0.935 ± 0.011             | 0.791 ± 0.013     | 0.260 ± 0.022    | **0.879 ± 0.010**         |
> | Without Adaptive Optimization              | 0.955 ± 0.003             | 0.752 ± 0.005     | 0.279 ± 0.006    | 0.869 ± 0.013             |
> | Without RL Decision Agent & Adaptive Optimization | 0.818 ± 0.021       | 0.737 ± 0.009     | 0.200 ± 0.022    | 0.768 ± 0.020             |
> | Replaced by Human-Designed RL              | 0.900 ± 0.014             | 0.776 ± 0.009     | 0.175 ± 0.018    | 0.759 ± 0.015             |
>
> In Appendix Section L.2, we performed ablation studies where we specifically removed each of the reviewer agents, combinations of them, and all of them to see whether fewer agents can achieve similar performance. As seen in Table A5, the **removal of any reviewer agents resulted in decreased model performance, justifying the requirement for all 3 Reviewer agents.**
>
> **Table A5: Ablation studies of the AI-simulated Peer Review module: property scores of antimicrobial peptides generated by MAC-AMP and its variants. RA1 = GPT-5,  RA2 = Perplexity, RA3 = Gemini 2.5**
>
> | Model            | Antibacterial Activity (↑) | AMP Likelihood (↑) | Toxicity (↓) | Structural Reliability (↑) |
> |-----------------|----------------------------|------------------|-------------|----------------------------|
> | MAC-AMP          | 0.943 ± 0.008             | 0.797 ± 0.012    | 0.154 ± 0.008 | 0.873 ± 0.009             |
> | w/o RA3          | 0.857 ± 0.016             | 0.840 ± 0.008    | 0.133 ± 0.023 | 0.731 ± 0.017             |
> | w/o RA2          | 0.884 ± 0.015             | 0.780 ± 0.008    | 0.164 ± 0.023 | 0.734 ± 0.021             |
> | w/o RA1          | 0.953 ± 0.010             | 0.742 ± 0.016    | 0.181 ± 0.019 | 0.783 ± 0.017             |
> | w/o RA2 + RA3    | 0.870 ± 0.013             | 0.806 ± 0.006    | 0.130 ± 0.015 | 0.740 ± 0.011             |
> | w/o RA1 + RA3    | 0.979 ± 0.010             | 0.700 ± 0.019    | 0.185 ± 0.019 | 0.758 ± 0.015             |
> | w/o RA1 + RA2    | 0.938 ± 0.011             | 0.795 ± 0.008    | 0.195 ± 0.013 | 0.862 ± 0.013             |
> | w/o All Review   | 0.825 ± 0.016             | 0.627 ± 0.014    | 0.448 ± 0.051 | 0.831 ± 0.012             |

---

> ### Author Response · Authors · 2025-11-26
>
> **Q1 – Sensitivity of MAC-AMP to Reviewer Agent LLMs**
>
> Thank you for suggesting that we validate the choice of our Reviewer agents in the framework. To ensure that the LLMs used were the optimal choice, we tested replacing the Reviewer and Area Chair agents with Qwen 2.5 7B-instruct and/or LLaMA 3.1 8B, which are locally deployed small models. As shown in Table A7, MAC-AMP achieves the best overall balance across all metrics, excelling in toxicity and AMP likelihood while maintaining competitive antibacterial activity and structural reliability. Variants with higher antibacterial activity or structural reliability incur substantially higher toxicity (2–3×) and lower AMP likelihood. By jointly prioritizing all objectives, MAC-AMP provides the most effective overall performance. Although it has the highest API and computational costs (except peak memory) as seen in Table R1 (Figure A10 in manuscript), performance is prioritized over efficiency, keeping costs acceptable for the intended use. More details can be found in Appendix Section M.3.
>
> **Table A7: Substitution analyses on the AI-simulated Peer Review Module Agents: property scores of antimicrobial peptides generated by MAC-AMP and its variants.RA\_AC\_Llama: all Reviewer agents (RA) and the Area Chair (AC) agent are replaced with Llama 3.1 8B, RA\_AC\_Qwen: all RA and the AC agent are replaced with Qwen 2.5 7B-instruct, RA\_API\_AC\_Qwen: only the AC agent was replaced with Qwen, RA\_Qwen\_AC\_API: only the RA are replaced with Qwen.**
>
> | Model             | Antibacterial Activity (↑) | AMP Likelihood (↑) | Toxicity (↓) | Structural Reliability (↑) |
> |------------------|----------------------------|------------------|-------------|----------------------------|
> | MAC-AMP           | 0.943 ± 0.008             | 0.797 ± 0.012    | 0.154 ± 0.008 | 0.873 ± 0.009             |
> | RA_AC_Llama       | 0.944 ± 0.007             | 0.713 ± 0.012    | 0.201 ± 0.020 | 0.896 ± 0.012             |
> | RA_AC_Qwen        | 0.949 ± 0.009             | 0.683 ± 0.010    | 0.308 ± 0.013 | 0.913 ± 0.016             |
> | RA_API_AC_Qwen    | 0.865 ± 0.014             | 0.783 ± 0.011    | 0.206 ± 0.013 | 0.863 ± 0.007             |
> | RA_Qwen_AC_API    | 0.835 ± 0.023             | 0.734 ± 0.017    | 0.211 ± 0.030 | 0.854 ± 0.015             |
>
> **Table R1: Substitution analyses on the AI-simulated Peer Review Module Agents: computational costs of antimicrobial peptides generated by MAC-AMP and its variants. RA\_AC\_Llama: all Reviewer agents (RA) and the Area Chair (AC) agent are replaced with Llama 3.1 8B, RA\_AC\_Qwen: all RA and the AC agent are replaced with Qwen 2.5 7B-instruct, RA\_API\_AC\_Qwen: only the AC agent was replaced with Qwen, RA\_Qwen\_AC\_API: only the RA are replaced with Qwen.**
>
> | Model Variant           | Total GPU Hours | Peak Memory (MB) | Total API Calls | RL Module API Call (% of Total) | Reviewer Module API Call (% of Total) | Total Cost (USD) | OpenAI Cost (% of Total) | Perplexity Cost (% of Total) | GenAI Cost (% of Total)  |
> |------------------------|----------------|-----------------|----------------|---------------------------------|--------------------------------------|----------------|-------------------------|-----------------------------|------------------------|
> | Reviewer_AC_llama       | 19.92          | 14757           | 155            | 100.00                          | 0.00                                 | 5.02           | 100.00                  | 0.00                        | 0.00                   |
> | Reviewer_AC_Qwen        | 23.43          | 14219           | 151            | 100.00                          | 0.00                                 | 4.76           | 100.00                  | 0.00                        | 0.00                   |
> | Reviewer_API_AC_Qwen    | 31.50          | 14217           | 685            | 22.63                           | 77.37                                | 26.36          | 75.42                   | 8.19                        | 16.39                  |
> | Reviewer_Qwen_AC_API    | 26.54          | 14207           | 334            | 45.81                           | 54.19                                | 15.01          | 100.00                  | 0.00                        | 0.00                   |
> | MAC-AMP                 | 47.61          | 9106            | 853            | 15.12                           | 84.88                                | 36.56          | 82.69                   | 5.61                        | 11.71                  |

---

> ### Author Response · Authors · 2025-11-26
>
> **Q2 – Validity of Tag Weights in AI-simulated Peer Review module**
>
> Thank you for pointing out the importance of ensuring that appropriate lexicon weights are chosen in the AI-simulated peer review module.  In MAC-AMP, the lexicon weights are determined by the agent itself during a preparatory meeting with a human expert that occurs prior to training and establishes the injectable knowledge for the model.
> To ensure correct choices, substitution analyses were performed to evaluate the change in performance metrics when the lexicon weights decided by the model are offset 0.1. As shown in Table A10, decreasing the lexicon weights by 0.1 resulted in a slight negative impact on performance, whereas increasing the weights by 0.1 caused a steep decline. Overall, the **weights determined during the human expert–agent preparatory meeting remain reliable and efficient.** This analysis has been added to the revised manuscript in Appendix M.6.
>
> **Table A10: Substitution analyses on the Lexicon Weights: property scores of antimicrobial peptides generated by MAC-AMP and its variants**
>
> | Model                  | Antibacterial Activity (↑) | AMP Likelihood (↑) | Toxicity (↓) | Structural Reliability (↑) |
> |------------------------|----------------------------|------------------|-------------|----------------------------|
> | MAC-AMP                | 0.943 ± 0.008             | 0.797 ± 0.012    | 0.154 ± 0.008 | 0.873 ± 0.009             |
> | Lexicon Weight + 0.1 | 0.878 ± 0.023             | 0.794 ± 0.009    | 0.156 ± 0.012 | 0.836 ± 0.013             |
> | Lexicon Weight - 0.1 | 0.938 ± 0.019             | 0.804 ± 0.006    | 0.171 ± 0.012 | 0.871 ± 0.013             |
>
> **Q3/Q5 – Clarification of Reward Hacking/Feedback Collapse**
>
> We fully agree with the reviewer that reward hacking and feedback collapse are real risks during deep learning model development. When designing MAC-AMP, we had thought of this issue and had implemented the following designs in place:
> - Only the peptide generation policy is updated; property predictors and LLM reviewers are kept fixed.
> - Reward is anchored in multi-source biological signals: MIC regressor & MACREL outputs (S) and toxicity, structural reliability, physicochemical properties, Foldseek similarity (V).
> - Training uses a three-stage, short-horizon scheme with stage-wise reward redesign to prevent long-horizon self-reinforcement.
> - Evaluation safeguards: Multiple LLM reviewers + property predictors aggregated via a meta-score with conflict penalty.
> - Reward safeguards: Multi-objective design with stage-wise logging ensures no single metric dominates; reweighting prevents over-optimization.
> - Constraint safeguards: Foldseek similarity and physicochemical properties suppress trivial or implausible solutions.
>
> To confirm this, we also analyzed and provided a three-stage reward–episodes learning curve in Figure A1, showing the evolution of total reward and its components, Sa, Sb, and Sc during training. Stage-specific patterns are observed: early-stage improvements focus on Sb (AMP-likeness), mid-stage increases target Sc (average meta score representing multi-agent assessments of toxicity, structural reliability, etc.), and late-stage optimization emphasizes Sa (antibacterial activity). **Across all stages, total reward rises steadily without evidence of reward hacking or feedback collapse.** More details can be found in Appendix D.
>
> **Q4 – Novelty of Generated Peptides**
>
> Although novelty is one of the objectives that MAC-AMP optimizes for during peptide generation, it is good to validate the novelty of the generated peptides externally. To do so, we quantified sequence similarity of the 90 anti-*E. coli* generated AMPs to the training dataset using the Needleman–Wunsch global alignment algorithm, normalizing scores by the length of the longer peptide. For each peptide, both the maximum and average similarity across all database sequences were computed. The results show a **maximum similarity of 84.6% (~15% sequence divergence) and an average similarity of 27%, indicating substantial sequence-level novelty** (Figure A4). Internal pairwise alignments among the 90 generated peptides also reveal consistently low similarity, demonstrating that MAC-AMP produces a diverse set of peptides rather than minor variations of a few motifs (Figure A5). Full details are provided in Appendix H.

---

> ### Author Response · Authors · 2025-11-26
>
> **Q6 – Computational Costs and Scalability with Peptide Length or Number of Objectives**
>
> Thank you for suggesting that we report the computational costs associated with MAC-AMP and its scalability. First, to train MAC-AMP for *E. coli* AMP generation, we used 47.61 GPU hours, made 853 API calls, consumed 9106 MB of peak memory, and incurred a total API token cost of $36.56 USD.
> We also performed analyses to see how the computational costs scales with peptide length or number of objectives.
> In MAC-AMP, the default maximum peptide length is 32, with variants of 20, 26, and 39 tested. As shown in Table R2 (Figure A9 in manuscript), differences in length have a negligible impact on computational costs. This analysis has been added to the revised manuscript in Appendix M.2.
>
> **Table R2: Substitution analyses on the Maximum Generated Peptide Lengths: computational costs of antimicrobial peptides generated by MAC-AMP and its variants**
> | Model Variant      | Total GPU Hours | Peak Memory (MB) | Total API Calls | RL Module API Calls (% of Total) | Reviewer Module API Calls (% of Total) | Total Cost (USD) | OpenAI Cost (% of Total) | Perplexity Cost (% of Total) | GenAI Cost (% of Total) |
> |--------------------|----------------|------------------|----------------|---------------------------------|-----------------------------------------|------------------|---------------------------|-------------------------------|--------------------------|
> | len_20             | 46.41          | 9035             | 845            | 14.32                           | 85.68                                   | 36.19            | 82.90                    | 5.69                          | 11.41                   |
> | len_26             | 46.83          | 9087             | 857            | 15.52                           | 84.48                                   | 37.28            | 82.40                    | 5.61                          | 11.99                   |
> | MAC-AMP_len_32     | 47.61          | 9106             | 853            | 15.12                           | 84.88                                   | 36.56            | 82.69                    | 5.61                          | 11.71                   |
> | len_39             | 48.15          | 9121             | 851            | 14.92                           | 85.08                                   | 37.34            | 82.32                    | 5.87                          | 11.81                   |

---

> ### Author Response · Authors · 2025-11-26
>
> We also conducted ablation studies on the Property Prediction module by removing individual objectives: Toxicity (Va), Structural Reliability (Vb), and AMP Likelihood (Sb), and comparing performance to the full MAC-AMP. As seen in Table R3 (Figure A7 in manuscript) and Table A4, while removing objectives slightly reduces computational costs and can improve the remaining optimized metrics, overall multi-objective performance declines. Additionally, omitting more than two objectives destabilizes the AI-based peer review module, as insufficient information can cause the RL feedback loop to terminate prematurely. Full details are found in Appendix section L.1.
>
> **Table R3: Ablation studies of the Property Prediction module: computational costs of antimicrobial peptides generated by MAC-AMP and its variants**
> | Model Variant   | Total GPU Hours | Peak Memory (MB) | Total API Calls | RL Module API Calls (% of Total) | Reviewer Module API Calls (% of Total) | Total Cost (USD) | OpenAI Cost (% of Total) | Perplexity Cost (% of Total) | GenAI Cost (% of Total) |
> |-----------------|------------------|-------------------|------------------|-----------------------------------|------------------------------------------|-------------------|----------------------------|-------------------------------|---------------------------|
> | MAC-AMP         | 47.61            | 9106              | 853              | 15.12                             | 84.88                                    | 36.56             | 82.69                      | 5.61                         | 11.71                    |
> | Drop_Vb         | 51.14            | 8831              | 857              | 15.52                             | 84.48                                    | 37.22             | 81.46                      | 5.86                         | 12.68                    |
> | Drop_Va         | 51.13            | 8763              | 863              | 16.11                             | 83.89                                    | 36.13             | 82.15                      | 5.78                         | 12.07                    |
> | Drop_Sb         | 53.75            | 8929              | 859              | 15.72                             | 84.28                                    | 37.54             | 81.70                      | 5.75                         | 12.55                    |
> | Drop_VaVb       | 44.55            | 8524              | 818              | 16.38                             | 83.62                                    | 34.10             | 83.34                      | 5.40                         | 11.26                    |
> | Drop_VbSb       | 38.81            | 8643              | 733              | 14.87                             | 85.13                                    | 31.31             | 81.83                      | 5.84                         | 12.33                    |
> | Drop_VaSb       | 54.58            | 8591              | 877              | 18.36                             | 81.64                                    | 35.66             | 84.32                      | 5.05                         | 10.63                    |
> | Drop_VaVbSb     | 30.39            | 8437              | 649              | 15.56                             | 84.44                                    | 26.69             | 82.47                      | 5.55                         | 11.99                    |
>
> **Table A4: Ablation studies of the Property Prediction module: property scores of antimicrobial peptides generated by MAC-AMP and its variants. Vb = Structural Reliability model (OmegaFold), Va = Toxicity model (ToxinPred 3.0), Sb = AMP Likelihood model (Macrel)**
> | Model              | Antibacterial Activity (↑) | AMP Likelihood (↑) | Toxicity (↓)     | Structural Reliability (↑) |
> |--------------------|----------------------------|---------------------|-------------------|-----------------------------|
> | MAC-AMP            | 0.943 ± 0.008              | 0.797 ± 0.012       | 0.154 ± 0.008     | 0.873 ± 0.009               |
> | Drop_Vb            | 0.904 ± 0.028              | 0.799 ± 0.009       | 0.157 ± 0.021     | 0.818 ± 0.012               |
> | Drop_Va            | 0.946 ± 0.045              | 0.782 ± 0.016       | 0.231 ± 0.028     | 0.799 ± 0.017               |
> | Drop_Sb            | 0.923 ± 0.037              | 0.742 ± 0.006       | 0.164 ± 0.010     | 0.830 ± 0.014               |
> | Drop_Va_Vb         | 0.904 ± 0.043              | 0.764 ± 0.023       | 0.183 ± 0.016     | 0.758 ± 0.015               |
> | Drop_Vb_Sb         | 0.889 ± 0.040              | 0.745 ± 0.035       | 0.134 ± 0.087     | 0.763 ± 0.017               |
> | Drop_Va_Sb         | 0.933 ± 0.021              | 0.735 ± 0.010       | 0.255 ± 0.024     | 0.836 ± 0.019               |
> | Drop_Va_Vb_Sb      | 0.866 ± 0.044              | 0.729 ± 0.024       | 0.212 ± 0.045     | 0.742 ± 0.031               |

---

### Author Response · Authors · 2025-12-03

Dear Area Chair,

We sincerely appreciate and recognize that this year presented an unusually heavy load and pressure, and we are sincerely grateful for your time, careful evaluation, and thoughtful consideration. To facilitate your decision and provide a clear overview, we summarize the following

(1)	core motivation of the paper

(2)	the reviewers’ assessments

(3)	additional analyses completed during rebuttal to address the reviewers’ concerns

## (1) Core Motivation and Novelties of Our Paper

Our work introduces **MAC-AMP**, the first framework explicitly designed for **closed-loop, multi-agent AMP design**. Existing AI-driven AMP design approaches struggle with unstable multi-objective optimization, opaque evaluation signals, and difficulty translating outputs into reproducible training signals. MAC-AMP addresses these gaps by integrating multi-agent consensus, structured peer review, reinforcement learning refinement, and closed-loop peptide generation into a single, end-to-end system. These innovations enable stable optimization, explainable evaluation, and continuous design improvement.

The key contributions of MAC-AMP include:

1.	Fully autonomous, closed-loop multi-agent system

2.	Stepwise explainability and auditability, overcoming black-box limitations

3.	Native support for multi-objective AMP design

4.	Domain-agnostic framework that supports transferability beyond AMP generation

## (2) Reviewer Ratings and Assessments
| Reviewer | Original Rating | Comments |
|----------|-----------------|----------|
| vtM8 | 6 | Highlighted the innovation of MAC-AMP in integrating multi-agent LLM collaboration with reinforcement learning for AMP design. Noted strengths include transparent logging, explainable reward signals, ablation studies, and consistent gains across multiple bacterial targets. Suggested clarifications on sensitivity to specific LLMs, validation of lexicon-tagging weights, potential feedback collapse, novelty beyond Foldseek similarity, reward variance, and computational costs. |
| yaxY | 4 | Appreciated the conceptual novelty and clear modular design. Raised concerns on limited experimental validation, reliance on PPO and pre-existing peptide generators, lack of quantitative ablations per reviewer or Area Chair, unclear computational cost/scalability, and untested cross-domain generalization. |
| xSXc | 2 | Praised the novel closed-loop multi-agent framework that compiles structured peer-review consensus into PPO-executable rewards for interpretable multi-objective AMP optimization, with consistent gains across activity, toxicity, and structural reliability metrics supported by clear architecture and ablation studies. Pointed out reliance on external tools without robustness validation, limited verification of structural metrics, potential ranking-selection bias, insufficient sensitivity analyses, and release of data and code. |
| yJGN | 6 | Emphasized the potential impact of integrating multi-agent collaboration with RL and structured peer review. Noted modularity for adaptation to other molecular/protein design tasks. Suggested estimating computational cost and evaluating performance with smaller open-source LLMs. |

Overall, the reviewers expressed strong support for our algorithm and approach, providing constructive and helpful feedback. Their comments primarily focused on requests for clearer explanations, along with suggestions for additional analyses. The ratings were generally positive. All points raised have been addressed through detailed clarifications and substantial new experiments, and these updates have been incorporated into the revised manuscript.

---

> ### Author Response · Authors · 2025-12-03
>
> ## (3) Additional Analyses Completed
>
> Based on the review discussion, here's a table summarizing the new experiments added and concerns addressed:
>
> | New experiments added | Concerns addressed |
> |------------------------|---------------------|
> | **MD simulations** validating OmegaFold pLDDT as structural reliability proxy; mean RMSD values 2–4 Å confirm stability | W1-vtM8, W2-xSXc, W3-xSXc: Property predictor accuracy; OmegaFold validation for short peptides |
> | **Toxicity predictor comparison**: ToxinPred 3.0 vs CAPTP on independent test set; ToxinPred superior (0.828 vs 0.770 accuracy) | W1-vtM8, W2-xSXc: Predictor quality constraints |
> | **MIC predictor comparison**: MAC-AMP predictor vs APEX; MAC-AMP superior (R²: 0.572 vs 0.546) | W1-vtM8, W2-xSXc: MIC prediction reliability |
> | **Cross-species validation**: APEX external testing on 90 peptides across ESKAPE pathogens; 40% broad-spectrum activity | W2-vtM8: Cross-species generalization reasoning |
> | **Motif analysis**: Conserved motifs (e.g., KFLKGA, WLLGKW) linked to mechanistic AMP activity identified in broad-spectrum peptides | W2-vtM8: Mechanistic basis for generalization |
> | **Ablation studies on reviewer agents**: removal of any agent (GPT-5, Perplexity, Gemini 2.5) decreases performance (already in initial submission) | W3-vtM8, W3-yaxY: Validation of agent number and reward design |
> | **Ablation studies on RL refinement module**: removal of components decreases performance (already in initial submission) | W3-vtM8: Validation of reward design complexity |
> | **Substitution analyses on reviewer LLMs**: replaced with Llama 3.1 8B and Qwen 2.5 7B; API versions maintain best balance | Q1-vtM8, W3-yaxY, W1-yJGN: LLM choice sensitivity; computational costs |
> | **Lexicon weight validation**: ±0.1 offsets demonstrate optimal weights | Q2-vtM8: Tag weight validity |
> | **Reward-episode learning curves**: three-stage analysis shows no reward hacking or feedback collapse | Q3/Q5-vtM8: Reward hacking prevention |
> | **Novelty quantification**: Needleman–Wunsch alignment; max 84.6% similarity to training data, average 27% | Q4-vtM8: Sequence-level novelty verification |
> | **Computational cost vs peptide length** (20, 26, 32, 39 aa): negligible impact | Q6-vtM8: Scalability with peptide length |
> | **Computational cost vs objectives**: ablations removing Va, Vb, Sb reveal trade-offs | Q6-vtM8, W4-xSXc: Scalability with objectives |
> | **Computational cost vs RL epochs** (10, 15, 20): modest increases observed | W4-xSXc, W2-yJGN: RL training scalability |
> | **Computational cost vs generator size** (Llama 1B, 8B, 85M): runtime unchanged | W4-xSXc: Multi-agent scalability |
> | **Physicochemical property comparison**: 2000 random peptides vs training data and baselines (not top-30) | W3-xSXc: Ranking bias mitigation; fair comparison |
> | **Domain transfer experiment**: ToTTo table-to-text generation task; improvements over T5 baseline | W5-yaxY: Cross-domain transferability proof |
> | **Complete cost breakdown**: 47.61 GPU hours, 853 API calls, \$36.56 USD; detailed by provider | W1-yJGN, W2-yJGN, W4-xSXc: Cost transparency and efficiency |
> | **Code and data release**: GitHub repository with conda environment, pinned dependencies, installation instructions | W1-xSXc, Q1-xSXc, Q2-xSXc: Reproducibility |
>
> We thank the reviewers for their thoughtful feedback and have conducted extensive additional experiments to address all raised concerns. We hope this summary is helpful for your evaluation. Thank you for your time and consideration, and we would be glad to provide any further clarification as needed.
>
> **Sincerely,**
> *The Authors*

---

### Meta-Review · Area_Chair_udE1 · 2026-01-06

**Summary:**

This study proposes  **MAC-AMP**, a closed-loop multi-agent collaboration (MAC) system for multi-objective AMP design task. The key innovation lies in converting structured multi-agent textual peer review consensus into executable reinforcement learning reward signals, enabling autonomous and interpretable optimization. Extensive performance evaluation across multiple bacterial targets shows MAC-AMP consistently outperforms SOTA baselines.

Reviewers raised major concerns in three aspects:

**Major concern 1: Evaluation is highly dependent on external tools (raised by Reviewer vtM8, yaxY, xSXc )**

The AMP design quality of MAC-AMP is tightly coupled to the accuracy of third-party property predictors (ToxinPred 3.0, OmegaFold, Macrel). No systematic robustness, or cross-validation (wet-lab experiments) is provided.

**Major concern 2: Insufficient ablation study (raised by Reviewer vtM8, yaxY, xSXc)**

More experimental validation is required for reward design and agent number, RL refinement module, and Substitution analyses on reviewer LLMs.

**Major concern 3: Reproducibility of the study (raised by Reviewer xSXc)**

Code and data used in this study should be released.

**Reviewer Concerns:**

During the rebuttal, the authors have addressed nearly all of the reviewers’ major concerns by supplementing additional experimental validations and analyses.

**Major concern 1: Evaluation is highly dependent on external tools (raised by Reviewer vtM8, yaxY, xSXc )**

To verify the reliability of the evaluation pipeline, the authors provided supplementary results generated using state-of-the-art (SOTA) property prediction methods, in addition to cross-validation via physics-based molecular dynamics (MD) simulations. In my view, this enhanced evaluation is now sufficiently rigorous and satisfying. Given that this is a computational lab, wet-lab experimental validation would likely impose an overly burdensome requirement.

**Major concern 2: Insufficient ablation study**

Extensive additional ablation studies and corresponding analyses to fully address the questions raised by the reviewers.

**Major concern 3: Transparency of the study**

The authors have released all relevant code and datasets.

**Reviewer Scores:**

Given that the authors have addressed all major concerns, I anticipate the reviewers will adjust their scores upward accordingly.

---

### Decision · Program_Chairs · 2026-01-26

Accept (Poster)